# Shared autonomic phenotype of long COVID and myalgic encephalomyelitis/chronic fatigue syndrome

Peter Novak[1,2¤]*, David M. Systrom[2,3], Alexandra Witte[1], Sadie P. Marciano[1], Donna Felsenstein[2,4], Jeff M. Milunsky[5], Aubrey Milunsky[5], Joel Krier[6], Mark C. Fishman[2,7]

1 Department of Neurology, Brigham and Women's Hospital, Boston, Massachusetts, United States of America, 2 Harvard University, Boston, Massachusetts, United States of America, 3 Department of Medicine, Pulmonary and Critical Care, Brigham and Women's Hospital, Boston, Massachusetts, United States of America, 4 Department of Infectious Diseases and Medicine, Mass General Brigham, Boston, Massachusetts, United States of America, 5 Center for Genetics, Cambridge, Massachusetts, United States of America, 6 Atrius Health, Boston, Massachusetts, United States of America, 7 Department of Stem Cell and Regenerative Biology, Boston Massachusetts, United States of America

¤ Current address: Department of Neurology, Mass General Brigham, Boston, Massachusetts, United States of America
* pnovak2@bwh.harvard.edu

## Abstract

### Introduction

Long COVID and myalgic encephalomyelitis/chronic fatigue syndrome (ME/CFS) are relatively common and disabling multisystem disorders that share overlapping features, including post-infectious onset and similar clinical manifestations such as brain fog, fatigue, muscle pain, and dysautonomia with orthostatic intolerance. These similarities suggest that Long COVID and ME/CFS may share common pathophysiological mechanisms, though the underlying mechanisms remain poorly understood, partly due to the difficulty in quantifying many of the symptoms.

### Materials and methods

This retrospective study evaluated Long COVID and pre-COVID ME/CFS patients who completed autonomic testing between 2018 and 2023 at the Brigham and Women's Faulkner Hospital Autonomic Laboratory. The evaluations included autonomic tests (Valsalva maneuver, deep breathing, tilt-table test, and sudomotor function) with capnography and transcranial Doppler monitoring of cerebral blood flow velocity (CBFv) in the middle cerebral artery, neuropathic assessment through skin biopsies for small fiber neuropathy (SFN), invasive cardiopulmonary exercise testing (ICPET), and laboratory analyses covering metabolic, inflammatory, autoimmune, and hormonal profiles.

**Data availability statement:** Data sharing will follow Data Sharing Policy of Mass General Brigham (MGB). In accordance with MGB policy [1], a Data Use Agreement (DUA) is required prior to any exchange of human subject data with an external party for research purposes. Principal Investigators (PIs) are responsible for ensuring that the appropriate approvals are in place before any data is shared. Specifically, the PI must consult with the Institutional Review Board (IRB) to determine the necessary type and level of IRB review applicable to the proposed research involving MGB data. No data may be shared until all institutional and regulatory requirements have been met. For questions related to IRB requirements, the IRB office may be contacted at IRB@mgb.org. For assistance with DUA templates or modifications, or to submit finalized documents, contact RMDUA@mgb.org. A copy of the signed DUA and signed attestation will be submitted to RMDUA@mgb.org as required. For more information, please refer to the MGB Data Sharing Policy: https://partnershealthcare.sharepoint.com/sites/phrm-Initiate/imcdc/Pages/Data-Use-Agreements-(DUAs).aspx 1: Data Sharing Policy at Mass General Brigham: 'https://partnershealthcare.sharepoint.com/sites/phrmInitiate/imcdc/Pages/Data-Use-Agreements-(DUAs).aspx'.

**Funding:** This work was funded by Mona Taliaferro/Bay Shore Recycling, The National Heart, Lung and Blood Institute (NHLBI - 1OT2HL156812-01) and FBRI LLC (2022A018462) to P. Novak. The funders had no role in study design, data collection and analysis, decision to publish, or preparation of the manuscript.

**Competing interests:** Dr. Novak is an advisor - independent Contractor for Dysimmune Diseases Foundation. Dr. Novak is a member of the Scientific Advisory Board of Endonovo Therapeutics. Dr. Novak received royalties from Oxford University Press.PN is a current or previous shareholder of Moderna, and Pfizer.

## Results

A total of 143 Long COVID and 170 ME/CFS patients were analyzed and compared to 73 healthy controls and 290 patients with hypermobile Ehlers-Danlos syndrome (hEDS). Tests revealed extensive similarities between Long COVID and ME/CFS, including reduced orthostatic CBFv (92%/88% in Long COVID/ME/CFS), mild-to-moderate widespread autonomic failure (95%/89%), presence of SFN (67%/53%), postural tachycardia syndrome (POTS) (22%/19%), neurogenic orthostatic hypotension (15%/15%) and preload failure (96%/92%, assessed in 25/66 Long COVID/ME/CFS). Patients with hEDS exhibited more severe peripheral neurodegeneration compared to the other groups. Laboratory tests did not distinguish between the conditions.

## Conclusion

Both Long COVID and ME/CFS demonstrate dysregulation in cerebrovascular blood flow, autonomic reflexes, and small fiber neuropathy, suggesting that these conditions may share a common underlying pathophysiology. However, differing distributions of findings in patients with hEDS raise the question of whether these conditions represent distinct but overlapping syndromes or reflect a shared underlying pathway. Further research is required to clarify the relationship between these conditions and the potential underlying pathophysiological mechanisms.

## Introduction

Postacute sequelae of SARS-CoV-2, also known as Long COVID, and myalgic encephalomyelitis/chronic fatigue syndrome (ME/CFS) are complex, multisystem disorders that significantly overlap in clinical presentation [1–6]. Approximately 10%−60% of individuals who have suffered for SARS-CoV-2 infection continue to experience or develop new symptoms consistent with Long COVID [7–9]. About 10%−80% of ME/CFS patients report preceding viral infection, mostly associated with the Epstein-Barr virus [10,11]. Both disorders are frequently disabling, and significantly impair daily functioning. Typical symptoms associated with both disorders include persistent fatigue, cognitive problems including brain fog, headaches, unrefreshing sleep, muscle and joint pain, post-exertional malaise, orthostatic intolerance, gastrointestinal symptoms and a variety of other symptoms [1,6]. While numerous abnormalities in immunologic, bioenergetics, and physiologic domains have been identified, the findings remain heterogenous, particularly within ME/CFS research and reliable biomarkers for both conditions have yet to be developed [4,11].

The pathophysiological mechanisms underlying both disorders remain incompletely understood. However, the overlapping symptomatology and frequent temporal association with antecedent infections raise the possibility of shared or converging biological pathways [4,6,12]. Emerging evidence increasingly implicates autonomic nervous system dysfunction as a common feature of both conditions [1].

Dysregulation of both the sympathetic and parasympathetic branches—manifesting as postural orthostatic tachycardia syndrome (POTS), orthostatic hypotension, autonomic failure and broader altered cardiac autonomic regulation has been implicated [13–15]. Impaired cerebral blood flow associated with or due to autonomic dysfunction may also be a common feature of both conditions [16]. Additionally, small fiber neuropathy (SFN) has been investigated as a potential underlying pathological substrate contributing to peripheral autonomic impairment [17,18].

In this study, we conducted a comparative analysis of Long COVID and ME/CFS using comprehensive assessment of cerebrovascular blood flow, autonomic reflexes, skin biopsies and metabolic, inflammatory, autoimmune, and hormonal profiles. These measurements were compared to historical healthy controls (excluding laboratory blood work data) and patients with hypermobile Ehlers-Danlos syndrome (hEDS), a heritable connective tissue disorder commonly associated with dysautonomia [19].

The underlying hypothesis tested was that Long COVID and ME/CFS share similar pathophysiology, and as a result, their key findings should be similar. By including hEDS as a disease control with a different etiology, we also evaluated whether the observed findings represent a common physiological response to illness or if they are specific to Long COVID and ME/CFS.

## Materials and methods

This retrospective, single-center study evaluated consecutive adult patients with a diagnosis of Long COVID, ME/CFS, and hEDS who underwent autonomic testing between January 1, 2018 and December 31, 2023 at the Brigham and Women's Faulkner Hospital Autonomic Laboratory, Boston, for evaluation of orthostatic intolerance. Clinical data were obtained from patients' electronic records. Data for research purposes were accessed on May 14, 2023 and on January 2, 2025.

### Standard protocol approvals, registrations, and patient consent

The study was approved by the Institutional Review Board of the Brigham and Women's Hospital, Harvard University, as a minimal-risk study, and the consent form signature was waived. Authors of the study had access to information that could identify individual participants during data collection.

### Clinical definitions

Orthostatic intolerance was defined as the presence or exacerbation of chronic (>6 months) symptoms attributable to cerebral hypoperfusion – such as lightheadedness, dizziness, dyspnea, brain fog, fatigue and visual disturbances – upon assuming an upright posture, with a partial or complete relief of symptoms upon recumbency. POTS was defined as a combination of orthostatic intolerance and an increment in heart rate ≥ 30 beats per minute for ages > 19 years and 40 beats per minute for ages 18–19 years without orthostatic hypotension during the tilt test [20]. Hypocapnic cerebral hypoperfusion (HYCH), was defined as a combination of orthostatic intolerance and reduced orthostatic cerebral blood flow velocity (CBFv) associated with hypocapnia (end-tidal $CO_2$ < 30 mmHg [21]), but without orthostatic tachycardia or orthostatic hypotension [22]. Orthostatic cerebral hypoperfusion syndrome (OCHOS) was defined by reduced orthostatic CBFv without orthostatic hypotension, orthostatic tachycardia and orthostatic hypocapnia [23].

### Inclusion and exclusion criteria

**Inclusion criteria.** Adults aged >18 years, both men and women, who completed autonomic testing and had a documented history of Long COVID and ME/CFS. Long COVID diagnosis was based on the following: 1) Evidence of previous SARS-CoV-2 infection—established by a history of acute illness (characterized by fever, cough and malaise) and confirmed by a positive SARS-CoV-2 test (either antigen or polymerase chain reaction). 2) Symptoms linked to Long COVID. Long COVID is a heterogenous condition, and at the time the study was performed, the exact diagnostic criteria

were still evolving [7]. Long COVID was defined as a constellation of persistent, relapsing or new symptoms after an acute infection [7,24]. These symptoms are brain fog, fatigue, smell/taste changes, post-exertional malaise, chronic cough, thirst, palpitations, dizziness, and gastrointestinal symptoms at variable combinations. The study included subjects who were infected during the pre-Delta era (before June 18, 2021), the Delta era (June 19, 2021 to December 18, 2021), and the Omicron era (after December 18, 2021) [25].

The diagnosis of ME/CFS was based on myalgic encephalomyelitis international consensus criteria (ME-ICC) [26] or newer National Academy of Medicine diagnostic criteria [27]. Key features of ME/CFS are chronic, severe, disabling fatigue, post-exertional malaise, brain fog, sleep disturbances such as unrefreshing sleep, variable pain syndromes, and orthostatic intolerance.

The Long COVID and ME/CFS subjects were compared to a healthy control group from our autonomic research database at the University of Massachusetts [28]. All healthy controls were asymptomatic and had normal responses to tilt in heart rate, blood pressure, CBFv and respiratory variables.

We compared the Long COVID and ME/CFS patients to those with hypermobile Ehlers-Danlos syndrome (hEDS), all of whom were evaluated in our laboratory using the same methodology. hEDS is a genetic disorder of connective tissue that is frequently associated with dysautonomia [29]. Unlike Long COVID and ME/CFS, there is no evidence implicating infection in the pathogenesis of hEDS. Key features of hEDS include joint hypermobility, variable pain syndromes, hyper-extensible skin, and autonomic dysfunction [29].

Diagnosis of hEDS was based on the Beighton-Villefranche criteria for patients seen prior to 2017) [30], and on the 2017 international criteria thereafter [31]. All hEDS diagnoses and were made by genetic specialists (JK, AM, or JM). The Beighton-Villefranche criteria do not distinguish between hEDS and hypermobile spectrum disorders, the latter being considered a milder form of hEDS [31]. Therefore, diagnoses made prior to 2017 were retrospectively confirmed using the updated international criteria.

**Exclusion criteria.** We excluded patients with hEDS who had a concurrent diagnosis of chronic fatigue [19] or met the diagnostic criteria for ME/CFS. Additionally, patients who did not complete or were unable to tolerate autonomic testing were excluded.

## Patient reported surveys

Patient-reported surveys were done as a part of autonomic testing. The Survey of Autonomic Symptoms (SAS) was used to assess the frequency and severity of autonomic symptoms [32]. The cutoff point > 7 in the SAS score was considered to be clinically significant. Sensory complaints were assessed by the self-reported Neuropathy Total Symptom Score-6 (NTSS-6) [33]. The NTSS-6 total score > 6 was considered as clinically significant. The pain for the last seven days was assessed using the 0–10 numerical rating pain scale (0 = no pain, 10 = worst imaginable pain), a part of the NIH Toolbox [34,35]. The scores ≤ 3 correspond to mild, scores 4–6 to moderate and scores ≥ 7 to severe pain. Central sensitization was assessed using the central sensitization inventory (CSI), a validated instrument used for evaluation of central sensitization [36]. The score ≥ 40 was used for the diagnosis of Central Sensitization Syndrome.

## Autonomic testing

All testing was performed following established standards and previously described in detail [22]. Medications that may affect autonomic function were discontinued for five half-lives or longer before the testing. Cardiovascular reflex tests included deep breathing, the Valsalva maneuver, and the tilt test. Deep breathing test was performed with inhalation and exhalation each equal to ten seconds which was repeated six times. Parasympathetic cardiovagal index was obtained as the average difference between expiratory and inspiratory heart rate. Valsalva maneuver was performed as a forced expiration at the expiratory pressure 40 mmHg for 15 seconds. The difference between baseline and end of the phase 2 in mean blood pressure was used as a sympathetic adrenergic index. Patients were tilted at 70 degrees for 10 minutes

following 10 minutes of supine rest. We described details of autonomic respiratory and cerebral blood flow measurements from the tilt previously [22].

Recorded signals included electrocardiogram, blood pressure, end-tidal $CO_2$, and CBFv in the middle cerebral artery using Transcranial Doppler. Blood pressure was obtained intermittently every minute by brachial sphygmomanometer using an automated monitor Welch Allyn CVSM 6400 Monitor (Skaneateles Falls, NY) and beat-to-beat using finger cuff the photoplethysmographic signal which was volume-clamped in the finger by servo control (Human NIBP Nano Interface MLA382, ADInstruments Inc., Colorado Springs, CO, USA and Human NIBP Nano Wrist Unit FMS910804, Finapress Medical Systems, Amsterdam, Netherlands). End-tidal $CO_2$ was obtained using Nonin Respsense Capnograph (Nonin Medical Inc. Plymouth, MN) by nasal cannula. A pulse oximeter (part of Welch Allyn monitor) was used to monitor the oxygen saturation throughout the testing.

The temporal acoustic window with a 2 MHz probe was used to acquire CBFv from the M1 segment of middle cerebral artery using a MultiDop T (Multigon, New York, NY) with an insonation depth between 45 and 65 mm. The transducer has been attached to the head using a head frame with a three-dimensional positioner. The depth and angle of insonation have been kept constant throughout the head-up tilt test. Signals were recorded using the PowerLab 16/35 data acquisition system with LabChart 8 software (ADInstruments Inc., Colorado Springs, CO, USA) and sampled at 400 Hz.

Electrochemical skin conductance (ESC) was used to measure the sudomotor function [37]. ESC correlates with loss of sweat gland nerve fibers and it is a reasonable proxy for sudomotor function [38].

Medical records were also searched for a history of invasive cardiopulmonary exercise testing (iCPET) [39] for evaluation of unexplained fatigue or dyspnea. iCPET was performed in a sitting position using a cycle ergometer as described in detail [40]. iCPET provides Fick cardiac output, right atrial pressure and previously iCPET showed impaired venous return and peripheral oxygen extraction in ME/CFS [18].

### Skin biopsies

Epidermal nerve fiber density (ENFD) and sweat gland nerve fiber density (SGNFD) were obtained at the proximal thigh 20 cm distal to the iliac spine and the calf 10 cm above the lateral malleolus using a 3-mm circular punch tool. Specimen processing including immunoperoxidase staining for the axonal marker PGP 9.5, and fiber counting was done at Therapath (New York, NY) using established standards [41].

### Criteria for small fiber neuropathy

SFN is defined as combination of clinical signs suggestive of small fiber dysfunction (pinprick and thermal sensory loss, allodynia, and hyperalgesia) and structural (obtained from skin biopsy) or functional (obtained from QSART or ESC) variables [37,41,42]. The following subtypes of SFN were assessed in this study: sensory SFN (abnormal ENFD, normal SGNFD), mixed SFN (abnormal ENFD and SGNFD), autonomic (normal ENFD, abnormal SGNFD), functional (abnormal electrochemical skin conductance (ESC)) and combined functional-morphological (at least one abnormal: ENFD, SGNFD, and ESC).

### Grading of autonomic tests

Test results were graded using the Quantitative Scale for Grading of Cardiovascular Autonomic Reflex Tests and Small Fibers from Skin Biopsies (QASAT). QASAT is an objective instrument for grading the severity of dysautonomia, small fiber neuropathy, and cerebral blood flow abnormalities that uses normative age and gender adjusted values as appropriate. Each domain (heart rate, blood pressure, cerebral blood flow, end-tidal $CO_2$) is analyzed separately, where a normal score equals to 0 and the score $\geq 0$ is abnormal. QASAT grading is defined as follows [22]:

Autonomic failure score ($QASAT_{af}$:

$$QASAT_{af} = QASAT_{cardiovagal} + QASAT_{adrenergic} + QASAT_{sudomotor}.$$

Cardiovagal failure score ($QASAT_{cardiovagal}$) is calculated from heart rate responses to deep breathing test. Adrenergic failure score ($QASAT_{adrenergic}$) is obtained as a sum of blood pressure responses to the Valsalva maneuver and head-up tilt scores. Sudomotor failure score ($QASAT_{sudomotor}$) is obtained from the ESC or QSART. The $QASAT_{af}$ range is 0–22; none (0), abnormality: mild (1–3), moderate (4–12), and severe (12–22). The additional QASAT ranges are defined as follows: cardiovagal failure: none (0), abnormality: mild (1), moderate (2) and severe (3); adrenergic failure – Valsalva maneuver: none (0), abnormality: mild (1), moderate (2) and severe (3); adrenergic failure – orthostatic hypotension: none (0), abnormality: mild (1), moderate (2–5) and severe (6–10); orthostatic tachycardia: none (0), abnormality: mild (1–2), moderate (3–5) and severe (6–10); sudomotor failure – ESC: none (0), abnormality: mild (1–2), moderate (3–4) and severe (5–6); sudomotor failure – QSART: none (0), abnormality: mild (1–2), moderate (3–6) and severe (7–8); ENFD: normal (0), abnormality: mild (1–2), moderate (3–6) and severe (7–8); SGNFD: normal (0), abnormality: mild (1–2), moderate (3–6) and severe (7–8); reduced orthostatic end-tidal $CO_2$: normal/none (0), abnormality: mild (1–2), moderate (3–5) and severe (6–10); reduced orthostatic CBFv: normal/none (0), abnormality: mild (1–2), moderate (3–5) and severe (6–10). Details of calculations and grading of the testing were published previously [43].

## Laboratory and inflammation markers

Both Long COVID and ME/CFS are postinfectious disorders were the autoimmunity and/or low grade inflammation may play a role [4]. Therefore, patient's charts were reviewed for laboratory blood evaluations conducted during routine clinical assessments. We specifically focused on inflammatory, autoimmune, and hormonal markers, as abnormalities in these have been reported in post-infectious disorders [3,44] including: high sensitivity C-reactive protein (normative value <=3 mg/L), tumor necrosis factor-alpha (TNF-α, <= 2.8 pg/mL), interleukin (IL) IL-6 (<7.1 pg/mL), IL-10 (<2 pg/mL), IL-1ß (<7.1 pg/mL), leptin (3.3–18.3 ng/mL), trisulfated heparin disaccharide (TS-HDS) antibody (<10000 titers) [45], fibroblast growth factor receptor 3 (FGFR3) antibody (<3000 titers) [45], acetylcholine receptor binding antibody (<=0.02 nmol/L), ganglionic acetylcholine receptor antibody (<= 0.02 nmol/L) [46], neuronal VGKC antibody (<= 0.02 nmol/L), calcium channel P/Q binding antibody (<= 0.02 nmol/L), myoglobin (<=71 ng/mL) [47] and human growth hormone (0.01–3.61 ng/mL) [47]. We also measured supine (70–750 pg/mL) and standing (200–1700 pg/mL) plasma norepinephrine levels, as these values are useful in assessing the hyperadrenergic form of POTS [20]. The systemic immune-inflammation index defined as neutrophils x platelets/lymphocytes was shown to be useful predictor marker in several malignances [48] was calculated as well.

TS-HDS and FGFR3 antibodies were obtained from Washington University School of Medicine (St. Louis, MO), remaining antibodies as well as TNF-α, IL-10, human growth hormone were obtained from Mayo Clinic laboratories (Rochester, MN). IL-6 was obtained at Mayo Clinic laboratories (Rochester, MN) or at BWH Clinical laboratories (Boston, MA). IL-1ß was obtained at Sunquest (Tucson, AZ). Leptin was obtained at Esoterix Endocrinology (Calabasas Hills, CA). Remaining of laboratory tests were obtained at BWH Clinical laboratories (Boston, MA) or at Quest Diagnostics (Secaucus, NJ).

## Statistical analysis

The continuous data were not normally distributed, and outliers were present. Additionally, the groups failed to meet the assumption of homogeneity of variances. We chose not to remove the outliers, as they may carry clinically important information, and their removal could significantly alter the data distribution. Therefore, we used the nonparametric Kruskal-Wallis test to compare the groups, as it does not assume a normal distribution and is relatively robust to outliers and heterogeneity of variances particularly in large datasets [49]. The effect of size was measured by epsilon squared. If an overall comparison was statistically significant, pairwise post hoc comparisons were performed using Dunn test [50] with the Benjamini-Hochberg

adjustment for multiple comparisons [51]. For categorical data, overall comparisons across groups were conducted using the Chi-squared as it does not assume normality [52]. If an overall comparison was statistically significant, pairwise post hoc comparisons were performed using the Fisher's Exact Test with Holm adjustment for multiple comparisons [53].

The effect of head-up tilt on hemodynamic variables was assessed using the linear mixed-effects models adjusted for supine baseline and with a random intercept with repeated-measures design [54]. The predictor variables were the diagnosis and position of the subjects (supine versus upright, minutes 1–10). Gender and age were covariates.

The relationship between lightheadedness during head-up tilt (absent versus present) and QASAT domains was evaluated using the binary logistic regression model.

A higher proportion of missing values was expected in the laboratory blood work, likely due to test ordering being at the discretion of the attending physicians. Ignoring missing data could affect the robustness of our findings. Assuming the data were missing at random, we conducted a sensitivity analysis to assess the potential impact. Specifically, we compared the results of the Kruskal–Wallis test for continuous variables across three scenarios: (1) complete-case analysis using the original dataset with missing data ignored, (2) imputation using the overall mean, and (3) imputation using the overall median.

The R software (www.r-project.org) was used for statistical analyses.

## Statistical power

The sample size for this study was determined based on a power analysis appropriate for detecting differences among four independent groups using the Kruskal-Wallis test, a nonparametric method suitable for non-normally distributed data. Assuming an alpha level of 0.05 and a desired statistical power of 0.80, the analysis utilized a logistic distribution to approximate the underlying data characteristics. For the primary outcome variable (total QASAT value), based on preliminary estimates of the mean and standard deviation, the minimal sample size required for each group was determined to be 37.2.

## Results

Of the total number of consecutive patients referred for autonomic testing with diagnoses of Long COVID (n = 166), ME/CFS (n = 203), and hEDS (n = 352), a subset was excluded due to incomplete or missing data (Fig 1). Ultimately, 143 patients with Long COVID and 170 with ME/CFS—predominantly younger women—were included in the study. These groups were compared to 73 healthy controls and 290 consecutive patients with hEDS (Table 1).

From Long COVID patients, 6 were of pre-Delta era, 10 of Delta era and remaining were Omicron era. All ME/CFS and hEDS patients were diagnosed before the onset of SARS-CoV-2 pandemics.

Long COVID and ME/CFS patients had similar age and gender, but Long COVID had a higher body mass index (BMI, p=0.006)) and shorter (p<0.001) symptom duration defined as the length of time since disease onset. Co-morbidities and medications were similar, but fibromyalgia (p=0.01), chronic Lyme disease (p=0.004), and the use of pressor medications (p<0.001) were more frequent in ME/CFS. Compared to Long COVID and ME/CFS, hEDS patients were younger (p<0.001), had more frequent mast cell activation syndrome (p<0.001), irritable bowel syndrome (p=0.03), more pain (p<0.01), headaches (p<0.00) and treatment with antihistamine medications (p<0.01). Laboratory evaluations that comprised of a spectrum of metabolic, hormonal, blood, inflammatory and autoimmune markers, available only in a subset of patients, were unrevealing, mostly within the normal range, and were similar between the Long COVID and ME/CFS and hEDS.

The sensitivity analysis showed that missing values did not significantly affect the robustness of our results.

## Symptoms

Long COVID and ME/CFS had a similar degree of complaints in the autonomic (total and subtotal scores on SAS), neuropathic (NTSS-6), pain, and central sensitization domains (Table 2). hEDS group had worse most of the SAS and NTSS-6 scores, and some central sensitization scores (pain, stiffness) (p=0.05-<0.001).

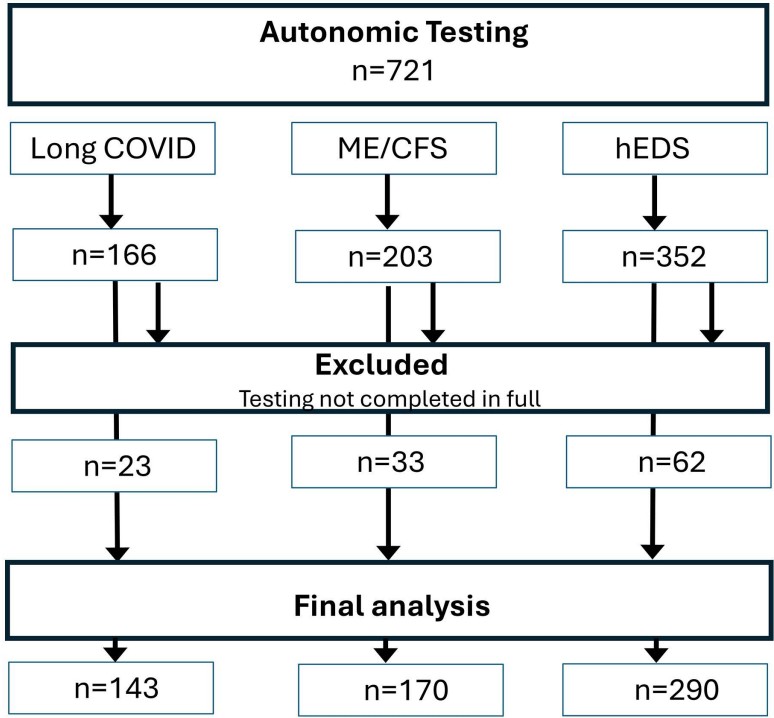

**Fig 1. Flow chart of the study.**

## Autonomic testing

Cardiovascular, cerebrovascular and respiratory variables at supine baseline and during the head-up tilt are shown in the Table 3 and Fig 2A–2D. Comparing all groups in the supine position, there was a significant difference in deep breathing, blood pressure response in the Valsalva maneuver, and ENFD ($p < 0.001$–0.03) but the difference was not significant in pair-wise comparisons. Furthermore, ESC was higher in hEDS compared to ME/CFS ($p = 0.008$), and Long COVID ($p < 0.001$). ENFD at the calf and SGNFD at the proximal thigh were higher in ME/CFS as compared to Long COVID ($p = 0.043$) and hEDS ($p = 0.049$). SGNFD was lower in hEDS compared to ME/CFS ($p = 0.007$). End-tidal $CO_2$ supine ($p = 0.004$) and orthostatic ($p < 0.001$) were lower in Long COVID.

The head-up tilt responses are shown in Fig 2A–2D as minute-by-minute profiles of cardiovascular, cerebrovascular and respiratory variables at rest and during the 10-minute head-up tilt. The patients with Long COVID and ME/CFS had similar heart rate and blood pressure increases during tilt (Fig 2A, 2B), reduced CBFv with a greater than 22% decline from baseline (Fig 2C), and end tidal $CO_2$ (Fig 2D). There was no difference for all tested variables between Long COVID and ME/CFS, except orthostatic end-tidal $CO_2$ was lower in the Long COVID group ($p < 0.001$). The hEDS group had a higher heart rate during tilt ($p < 0.03$) (Fig 2, Table 3), and higher orthostatic CBFv ($p < 0.001$) and lower cerebrovascular resistance as compared to Long COVID and ME/CFS. In all subjects, the oxygen saturation was within normal limits throughout the testing (range 96–99%).

Linear model showed significant effect of the diagnoses on the tilt responses for all hemodynamic variables ($p < 0.001$, Fig 2).

## QASAT

Overall comparisons showed abnormal QASAT scores (>0) in all domains in both groups, indicating mild-to-moderate dysautonomia. Fig 3A–3C and Tables 3 and 4 show absolute QASAT values, % of normalized scores, and frequency of

**Table 1. Demographic and baseline characteristics.**

| Variable | Control (n=73) | Long COVID (n=143) | ME/CFS (n=170) | hEDS (n=290) | P value Overall[a] | ε² | ME/CFS-Long COVID[b] | ME/CFS-hEDS[b] | Long COVID-hEDS[b] | Missing |
|---|---|---|---|---|---|---|---|---|---|---|
| Age, years | 39.84 (13.02) | 43.71 (13.23) | 44.45 (14.69) | 35.70 (12.33) | <0.001 | 0.081 | 0.879 | <0.001 | <0.001 | 0 |
| Gender, female % | 87.7 | 72.7 | 78.8 | 94.5 | <0.001 | | 0.233 | <0.001 | <0.001 | 0 |
| Race | | | | | | | | | | 0 |
| African American, % | 0.0 | 0.7 | 0.0 | 0.0 | | | | | | 0 |
| American Indian or Alaska Native, % | 0.0 | 0.0 | 0.8 | 0.7 | | | | | | 0 |
| Asian,% | 0.0 | 2.9 | 1.6 | 0.0 | | | | | | 0 |
| Multiracial, % | 0.0 | 0.7 | 0.8 | 0.7 | | | | | | 0 |
| White, % | 100.0 | 95.7 | 96.9 | 98.5 | 0.364 | | | | | 0 |
| BMI, m²/kg | 25.33 (4.87) | 27.81 (6.26) | 25.77 (6.07) | 25.60 (6.17) | <0.001 | 0.025 | 0.006 | 0.846 | <0.001 | 0 |
| Symptoms duration*, years | 0.76 (1.83) | 1.89 (0.89) | 10.22 (8.68) | 11.74 (8.07) | <0.001 | 0.53 | <0.001 | 0.015 | <0.001 | 0 |
| **Comorbid conditions** | | | | | | | | | | |
| Diabetes mellitus, % | 0.0 | 0.7 | 1.8 | 1.7 | 0.572 | | | | | 0 |
| Lyme disease, chronic, % | 0.0 | 0.7 | 9.4 | 3.4 | <0.001 | | 0.004 | 0.044 | 0.331 | 0 |
| Mast cell activation syndrome, % | 0.0 | 0.7 | 3.5 | 41.0 | <0.001 | | 0.392 | <0.001 | <0.001 | 0 |
| Hereditary alpha tryptasemia, % | 0.0 | 0.0 | 1.8 | 3.4 | 0.05 | | 0.890 | 0.89 | 0.174 | 0 |
| Depression, % | 0.0 | 52.6 | 62.4 | 70.5 | <0.001 | | 0.460 | 0.460 | 0.047 | 387 |
| Fibromyalgia, % | 0.0 | 8.2 | 25.0 | 26.1 | 0.001 | | 0.011 | 0.999 | 0.008 | 387 |
| Irritable bowel syndrome, % | 0.0 | 27.6 | 27.1 | 46.6 | <0.001 | | 0.999 | 0.031 | 0.031 | 386 |
| Anxiety, % | 0.0 | 60.2 | 52.9 | 73.9 | <0.001 | | 0.37 | 0.014 | 0.123 | 386 |
| Headaches, % | 0.0 | 53.1 | 49.4 | 80.7 | <0.001 | | 0.658 | <0.001 | <0.001 | 386 |
| **Current medication** | | | | | | | | | | |
| Anti-histamine, % | 0.0 | 46.2 | 42.9 | 60.7 | <0.001 | | 0.648 | <0.001 | 0.011 | 0 |
| Pain, % | 0.0 | 49.0 | 48.2 | 62.8 | <0.001 | | 0.910 | 0.008 | 0.014 | 0 |
| Pressor, % | 0.0 | 23.8 | 43.5 | 30.7 | <0.001 | | <0.001 | 0.013 | 0.142 | 0 |
| Psychiatric, % | 0.0 | 49.7 | 52.9 | 52.1 | <0.001 | | 0.999 | 0.999 | 0.999 | 0 |
| Hypertension, % | 0.0 | 13.3 | 12.4 | 6.6 | 0.002 | | 0.866 | 0.087 | 0.087 | 0 |
| Antitachycardic, % | 0.0 | 21.0 | 16.5 | 18.3 | 0.001 | | 0.934 | 0.999 | 0.999 | 0 |
| Gastrointestinal, % | 0.0 | 21.7 | 26.5 | 35.5 | <0.001 | | 0.356 | 0.099 | 0.012 | 0 |
| Immunomodulators, % | 0.0 | 2.1 | 2.9 | 5.2 | 0.103 | | 1 | 1 | 0.996 | 0 |
| **Laboratory evaluations** | | | | | | | | | | |
| C-reactive protein, high sensitivity normal ≤3 mg/L | | 2.14 (2.82) | 4.62 (7.95) | 3.14 (5.85) | 0.369 | 0.003 | | | | 407 |
| C-reactive protein high sensitivity, % abnormal | | 21.4 | 32.3 | 25.3 | 0.434 | | | | | 408 |
| Interleukin 6, normal <7.1 pg/mL | | 3.11 (2.04) | 3.28 (1.90) | 3.02 (1.48) | 0.464 | 0.003 | | | | 483 |
| Interleukin 6, % abnormal | | 3.3 | 5.9 | 2.7 | 0.735 | | | | | 485 |
| Interleukin 1b, normal <0.1 pg/mL | | 1.25 (5.26) | 1.82 (5.83) | 0.37 (1.09) | 0.504 | <0.001 | | | | 506 |
| Interleukin 1b, % abnormal | | 3.8 | 5.7 | 6.1 | 0.924 | | | | | 509 |
| Tumor necrosis factor alpha, normal ≤2.8 pg/mL | | 2.85 (3.12) | 2.85 (4.69) | 3.17 (4.64) | 0.316 | 0.004 | | | | 473 |
| Tumor necrosis factor alpha, % abnormal | | 19.4 | 15.4 | 22.7 | 0.656 | | | | | 476 |

*(Continued)*

| Variable | Control (n = 73) | Long COVID (n = 143) | ME/CFS (n = 170) | hEDS (n = 290) | P value Overall[a] | ε² | ME/CFS-Long COVID[b] | ME/CFS-hEDS[b] | Long COVID-hEDS[b] | Missing |
|---|---|---|---|---|---|---|---|---|---|---|
| Leptin, normal range = 3.3–18.3 ng/mL | | 10.44 (10.17) | 18.43 (15.03) | 12.13 (13.99) | 0.182 | 0.006 | | | | 517 |
| Leptin, % abnormal | | 0.16 (0.37) | 0.48 (0.51) | 0.22 (0.42) | 0.029 | | 0.168 | 0.999 | 0.16 | 518 |
| Tryptase, normal < 11.5 ng/mL | | 4.60 (2.65) | 4.65 (2.05) | 4.95 (3.61) | 0.604 | 0.002 | | | | 358 |
| Tryptase, % abnormal | | 4.3 | 3.0 | 8.8 | 0.246 | | | | | 366 |
| Voltage gated potassium channel complex antibody, normal ≤0.02 nmol/L, % | | 1.46 (10.32) | 0.00 (0.00) | 0.01 (0.09) | 0.244 | 0.005 | | | | 351 |
| Voltage gated potassium channel complex antibody, nmol/L, % abnormal | | 2.0 | 0.0 | 3.1 | 0.336 | | | | | 352 |
| Calcium channel P/Q antibody, normal ≤ 0.02 nmol/L | | 0.00 (0.01) | 0.00 (0.00) | 0.00 (0.01) | 0.507 | | | | | 352 |
| Calcium channel P/Q antibody, % abnormal | | 4.1 | 1.4 | 1.5 | 0.509 | | | | | 352 |
| Trisulfated heparin disaccharide antibody normal titer<10000 | | 8900.00 (15673.40) | 10700.00 (12728.36) | 7400.00 (11998.52) | 0.607 | <0.001 | | | | 503 |
| Trisulfated heparin disaccharide antibody, % abnormal | | 30.0 | 50.0 | 30.0 | | | | | | |
| Fibroblast growth factor receptor 3 antibody, normal titer < 3000 | | 310.00 (980.31) | 1970.00 (3211.11) | 896.47 (2015.03) | 0.228 | 0.005 | | | | 498 |
| Fibroblast growth factor receptor 3 antibody, % abnormal | | 11.1 | 20.0 | 17.6 | | | | | | 499 |
| Neutrophil, normal range = 1.8–7.7 K/µL | | 3.65 (1.38) | 4.18 (1.53) | 4.30 (1.37) | 0.032 | 0.011 | 0.138 | 0.028 | 0.319 | 442 |
| Neutrophil, % abnormal | | 0 | 0 | 0 | 0.624 | | | | | 442 |
| Lymphocyte, normal range = 1.0–4.8 K/µL | | 1.76 (0.70) | 1.94 (0.64) | 1.90 (0.62) | 0.264 | 0.011 | | | | 442 |
| Lymphocyte, % abnormal | | 0 | 0 | 0 | | | | | | |
| Neutrophil/Lymphocyte ratio | | 2.23 (1.00) | 2.48 (1.96) | 2.44 (0.90) | 0.218 | 0.005 | | | | 452 |
| Platelet, normal range = 150–400 K/µL | | 270.79 (66.39) | 268.53 (66.96) | 268.25 (70.19) | 0.864 | <0.001 | | | | 432 |
| Platelet, % abnormal | | 0 | 0 | 0 | 0.474 | | | | | 432 |
| Systemic inflammation index, normal ≤ 545 | | 600.66 (277.47) | 678.67 (585.30) | 643.60 (260.84) | 0.44 | <0.001 | | | | 452 |
| Systemic inflammation index, % abnormal | | 53.5 | 50.0 | 59.6 | | | | | | 452 |
| Norepinephrine supine, normal range = 70–750 pg/mL | | 565.17 (446.03) | 510.91 (277.63) | 514.91 (228.82) | 0.63 | <0.001 | | | | 499 |
| Norepinephrine supine, % abnormal | | 17.4 | 8.6 | 8.9 | 0.496 | | | | | 500 |
| Norepinephrine standing, normal 200–1700 pg/mL | | 757.90 (519.96) | 701.81 (376.35) | 689.68 (344.49) | 0.97 | <0.001 | | | | 507 |
| Norepinephrine standing, % abnormal | | 19.0 | 22.6 | 16.3 | 0.792 | | | | | 508 |
| Cortisol, normal range = 6.0–18.4 µg/dL | | 11.15 (6.58) | 10.60 (5.23) | 13.55 (6.70) | 0.534 | 0.002 | | | | 544 |
| Cortisol, % abnormal | | 0.19 (0.40) | 0.11 (0.32) | 0.12 (0.35) | 0.729 | | | | | 547 |
| ACTH, normal range = 7.2–63 pg/mL | | 12.39 (9.51) | 20.91 (25.18) | 17.88 (12.76) | 0.454 | 0.003 | | | | 572 |
| ACTH, % abnormal | | 0.43 (0.53) | 0.31 (0.48) | 0.17 (0.41) | 0.596 | | | | | 574 |
| Myoglobin normal ≤ 71 ng/mL | | 32.07 (16.97) | 29.79 (14.73) | 26.14 (11.65) | 0.052 | 0.01 | 0.517 | 0.7 | 0.624 | 526 |

*(Continued)*

**Table 1.** (Continued)

| Variable | Control (n=73) | Long COVID (n=143) | ME/CFS (n=170) | hEDS (n=290) | P value | | | | | Missing |
|---|---|---|---|---|---|---|---|---|---|---|
| | | | | | Overall[a] | ε² | ME/CFS-Long COVID[b] | ME/CFS-hEDS[b] | Long COVID-hEDS[b] | |
| Myoglobin, % abnormal | | 0.05 (0.23) | 0.03 (0.18) | 0.04 (0.19) | 0.941 | | | | | 527 |
| Ferritin, normal range=20–300 µg/L | | 104.28 (143.99) | 144.48 (215.89) | 71.33 (66.66) | 0.354 | 0.003 | | | | 544 |
| Ferritin, % abnormal | | 0.12 (0.33) | 0.12 (0.33) | 0.25 (0.45) | 0.509 | | | | | 548 |

*, Symptom duration was defined as the length of time since disease onset. Antitachycardic=adrenergic beta blockers, calcium channel blockers, ivabradine; Pressors=proamatine, fludrocortisone, pyridostigmine, droxydopa; % abnormal, Percentage of abnormal tests. Data are mean±sd. %, Prevalence of respective variable in percent. $ε^2$, Epsilon squared; [a]=Calculated using Kruskal-Wallis or chi-squared test as appropriate. [b]=Pairwise comparison calculated using Dunn or Fisher Exact test as appropriate.

abnormal findings. Both the Long COVID and ME/CFS groups had a worse QASAT$_{sudomotor}$ score compared to the hEDS group (Figs 3A, 2B).

Orthostatic lightheadedness was observed in >65% of patients, but orthostatic dyspnea was reported only in 21% of Long COVID patients, 37% of ME/CFS patients, and 28% of hEDS patients (Table 4). Tests revealed extensive similarities between Long COVID and ME/CFS, including reduced orthostatic CBFv (80%/92% Long COVID/ME/CFS), mild-to-moderate widespread autonomic failure (89%/95%), presence of SFN (63%/67%), postural tachycardia syndrome (32%/22%) and neurogenic orthostatic hypotension (12%/17%).

Small fiber neuropathy affected ~80–90% of patients using combined morphological and functional criteria. In Long COVID, QASAT$_{ENFD}$ was abnormal in 48.3% compared to 33.5% in ME/CFS (p=0.02), but QASAT$_{SGNFD}$ 27.8% was similar to ME/CFS 28.8%. The rate of SFN (from any biopsy) was similar between Lon COVID 67.2% and hEDS 63.3%,but was higher than in ME/CFS 52.6% (p=0.04).

**Invasive cardiopulmonary exercise testing (iCPET)**

iCPET was done in a sitting position, and the results were available in 25 Long COVID and 66 ME/CFS (Table 5). Unadjusted resting stroke volume (Long COVID vs. ME/CFS: p=0.01), exercise stroke volume (p=0.01), cardiac output (p=0.003), and oxygen uptake (p=0.001) were higher in Long COVID; however, these differences were no longer significant after adjusting for BMI (which was higher in Long COVID). Preload failure was detected in 96% of Long COVID and 92.4% of ME/CFS patients. Deconditioning was present in 64% of Long COVID and ME/CFS patients.

## Discussion

We report here central sensitization, cerebrovascular, dysautonomic, and neurodegenerative attributes of Long COVID and ME/CFS which are shared among the vast majority of patients with these disorders.

### Comparing Long COVID with ME/CFS

Long COVID and ME/CFS are associated with central sensitization and abnormalities in multiple domains including cerebral blood flow and respiratory dysregulation, small fiber neuropathy, and widespread autonomic failure. Table 6 provides a quantitative summary of main difference between Long COVID and ME/CFS.

### Central sensitization

Evidence of central sensitization was frequently observed in the majority of our patients. Central sensitization refers to the increase responsiveness of the nervous system to stimuli and is linked to abnormal interoception [55–57]. The features

**Table 2. Patient's reported outcome measures.**

**Survey of autonomic symptoms**

| Score | Control (n=73) | Long COVID (n=143) | ME/CFS (n=170) | hEDS (n=290) | P value Over-all[a] | ε² | ME/CFS-Long COVID [b] | ME/CFS-hEDS[b] | Long COVID-hEDS[b] | Missing[c] |
|---|---|---|---|---|---|---|---|---|---|---|
| Total | 0.08 (0.28) | 22.50 (10.02) | 23.42 (9.17) | 29.72 (9.67) | <0.001 | 0.367 | 0.879 | <0.001 | <0.001 | 0 |
| Orthostatic | 0.08 (0.28) | 3.55 (1.34) | 3.44 (1.21) | 4.08 (1.08) | <0.001 | 0.336 | 0.285 | <0.001 | <0.001 | 0 |
| Sudomotor | 0.00 (0.00) | 6.54 (4.54) | 6.68 (4.53) | 8.78 (4.51) | <0.001 | 0.303 | 0.854 | <0.001 | <0.001 | 0 |
| Vasomotor | 0.00 (0.00) | 4.53 (2.99) | 5.17 (3.01) | 6.48 (2.88) | <0.001 | 0.304 | 0.095 | <0.001 | <0.001 | 0 |
| Gastrointestinal | 0.00 (0.00) | 6.00 (3.66) | 6.38 (3.78) | 8.56 (3.76) | <0.001 | 0.322 | 0.402 | <0.001 | <0.001 | 0 |
| Urinary | 0.00 (0.00) | 1.37 (1.61) | 1.34 (1.56) | 1.76 (1.71) | <0.001 | 0.113 | 0.896 | 0.011 | 0.021 | 0 |

**Neuropathy total symptom score-6**

| Score | Control (n=73) | Long COVID (n=143) | ME/CFS (n=170) | hEDS (n=290) | P value Over-all[a] | ε² | ME/CFS-Long COVID [b] | ME/CFS-hEDS[b] | Long COVID-hEDS[b] | Missing[c] |
|---|---|---|---|---|---|---|---|---|---|---|
| Total score | 0.00 (0.00) | 9.59 (4.83) | 9.65 (5.13) | 11.56 (4.48) | <0.001 | 0.315 | 0.771 | <0.001 | <0.001 | 0 |
| Aching frequency | 0.00 (0.00) | 2.51 (0.71) | 2.49 (0.85) | 2.59 (0.74) | <0.001 | 0.345 | 0.772 | 0.318 | 0.254 | 0 |
| Aching intensity | 0.00 (0.00) | 1.78 (0.76) | 1.86 (0.86) | 2.11 (0.70) | <0.001 | 0.331 | 0.343 | 0.003 | <0.001 | 0 |
| Allodynia frequency | 0.00 (0.00) | 0.84 (1.12) | 1.04 (1.15) | 1.34 (1.02) | <0.001 | 0.157 | 0.879 | <0.001 | <0.001 | 0 |
| Allodynia intensity | 0.00 (0.00) | 0.71 (1.02) | 0.95 (1.12) | 1.42 (1.06) | <0.001 | 0.183 | 0.056 | <0.001 | <0.001 | 0 |
| Burning frequency | 0.00 (0.00) | 1.37 (1.24) | 1.32 (1.23) | 1.61 (1.09) | <0.001 | 0.156 | 0.702 | 0.014 | 0.049 | 0 |
| Burning intensity | 0.00 (0.00) | 1.11 (1.05) | 1.12 (1.10) | 1.54 (1.06) | <0.001 | 0.178 | 0.988 | <0.001 | <0.001 | 0 |
| Lancinating frequency | 0.00 (0.00) | 1.59 (1.16) | 1.55 (1.21) | 1.73 (1.04) | <0.001 | 0.184 | 0.79 | 0.175 | 0.282 | 0 |
| Lancinating intensity | 0.00 (0.00) | 1.50 (1.16) | 1.47 (1.20) | 1.74 (0.98) | <0.001 | 0.192 | 0.819 | 0.029 | 0.06 | 0 |
| Prickling frequency | 0.00 (0.00) | 1.95 (1.11) | 1.85 (1.10) | 2.04 (0.90) | <0.001 | 0.24 | 0.354 | 0.165 | 0.728 | 0 |
| Prickling intensity | 0.00 (0.00) | 1.27 (0.86) | 1.36 (0.91) | 1.68 (0.80) | <0.001 | 0.27 | 0.387 | <0.001 | <0.001 | 0 |
| Numbness frequency | 0.00 (0.00) | 1.83 (1.12) | 1.65 (1.24) | 1.94 (1.01) | <0.001 | 0.08 | 0.879 | <0.001 | <0.001 | 0 |
| Numbness intensity | 0.00 (0.00) | 1.29 (0.92) | 1.24 (1.01) | 1.66 (0.87) | <0.001 | 0.24 | 0.681 | <0.001 | <0.001 | 0 |

**Numerical rating pain scale**

| Score | Control (n=73) | Long COVID (n=143) | ME/CFS (n=170) | hEDS (n=290) | P value Over-all[a] | ε² | ME/CFS-Long COVID [b] | ME/CFS-hEDS[b] | Long COVID-hEDS[b] | Missing[c] |
|---|---|---|---|---|---|---|---|---|---|---|
| Score | 0.00 (0.00) | 2.71 (2.84) | 2.90 (2.69) | 4.37 (2.41) | <0.001 | 0.239 | 0.492 | <0.001 | <0.001 | 2 |

**Central sensitization inventory**

| Score | Control (n=73) | Long COVID (n=143) | ME/CFS (n=170) | hEDS (n=290) | P value Over-all[a] | ε² | ME/CFS-Long COVID [b] | ME/CFS-hEDS[b] | Long COVID-hEDS[b] | Missing[c] |
|---|---|---|---|---|---|---|---|---|---|---|
| Central sensitization syndrome, % | | 78.1 | 85.4 | 92.4 | 0.046 | 0.009 | 0.41 | 0.41 | 0.05 | 359 |
| CSI score | | 52.42 (15.01) | 54.28 (13.51) | 64.14 (15.40) | <0.001 | 0.043 | 0.486 | <0.001 | <0.001 | 359 |
| Tiredness after waking up | | 3.09 (0.93) | 3.42 (0.77) | 3.47 (0.77) | 0.008 | 0.016 | 0.019 | 0.675 | 0.017 | 355 |

*(Continued)*

**Survey of autonomic symptoms**

| Score | Control (n=73) | Long COVID (n=143) | ME/CFS (n=170) | hEDS (n=290) | P value | | | | | Missing[c] |
|---|---|---|---|---|---|---|---|---|---|---|
| | | | | | Over-all[a] | ε² | ME/CFS-Long COVID [b] | ME/CFS-hEDS[b] | Long COVID -hEDS[b] | |
| Muscle stiffness | | 2.76 (0.99) | 2.87 (1.10) | 3.23 (0.86) | 0.009 | 0.016 | 0.267 | 0.078 | 0.007 | 355 |
| Anxiety | | 1.53 (1.24) | 0.95 (0.94) | 1.44 (1.08) | 0.003 | 0.02 | 0.004 | 0.012 | 0.8 | 355 |
| Clenching teeth | | 1.92 (1.17) | 1.78 (1.42) | 2.27 (1.23) | 0.074 | 0.009 | | | | 355 |
| Diarrhea or constipation | | 2.41 (1.20) | 2.42 (1.29) | 3.11 (0.96) | <0.001 | 0.027 | 0.863 | <0.001 | <0.001 | 355 |
| Need help with daily activities | | 1.39 (1.27) | 1.69 (1.32) | 1.92 (1.28) | 0.028 | 0.013 | 0.186 | 0.252 | 0.026 | 355 |
| Sensitive to bright light | | 2.29 (1.30) | 2.61 (1.27) | 2.98 (1.16) | 0.002 | 0.020 | 0.092 | 0.101 | 0.002 | 355 |
| Fatigue | | 3.26 (1.04) | 3.51 (0.83) | 3.39 (0.93) | 0.3 | 0.004 | | | | 355 |
| Pain | | 1.79 (1.43) | 2.20 (1.35) | 2.97 (1.21) | <0.001 | 0.05 | 0.063 | <0.001 | <0.001 | 355 |
| Headaches | | 2.28 (1.20) | 2.31 (1.13) | 2.82 (0.80) | 0.007 | 0.016 | 0.994 | 0.009 | 0.013 | 355 |
| Urinary problems | | 0.68 (0.99) | 0.67 (0.91) | 1.14 (1.12) | 0.006 | 0.017 | 0.81 | 0.012 | 0.009 | 355 |
| Sleep problems | | 2.55 (1.15) | 2.67 (1.17) | 2.85 (1.10) | 0.272 | 0.004 | | | | 355 |
| Concentration problems | | 2.74 (0.92) | 2.77 (1.06) | 2.82 (0.91) | 0.771 | <0.001 | | | | 355 |
| Skin problems | | 2.21 (1.23) | 2.20 (1.30) | 2.79 (1.26) | 0.004 | 0.018 | 0.899 | 0.007 | 0.007 | 355 |
| Stress-related symptoms | | 2.88 (1.06) | 2.88 (1.11) | 2.88 (1.30) | 0.844 | 0.001 | | | | 355 |
| Sadness/depression | | 1.85 (1.16) | 1.67 (1.01) | 1.91 (1.21) | 0.44 | 0.003 | | | | 355 |
| Low energy | | 3.18 (0.84) | 3.48 (0.79) | 3.33 (0.81) | 0.024 | 0.012 | 0.487 | 0.692 | 0.748 | 355 |
| Neck and shoulder tension | | 2.78 (1.27) | 2.95 (1.07) | 3.47 (0.77) | <0.001 | 0.024 | 0.587 | 0.004 | <0.001 | 355 |
| Jaw pain | | 1.25 (1.33) | 1.52 (1.27) | 2.30 (1.23) | <0.001 | 0.042 | 0.153 | <0.001 | <0.001 | 355 |
| Smell supersensitivity | | 1.48 (1.36) | 1.71 (1.55) | 2.47 (1.44) | <0.001 | 0.03 | 0.314 | 0.003 | <0.001 | 355 |
| Frequent urination | | 2.02 (1.32) | 1.90 (1.40) | 2.09 (1.32) | 0.658 | <0.001 | | | | 355 |
| Restless legs | | 1.44 (1.17) | 1.59 (1.23) | 2.24 (1.23) | <0.001 | 0.028 | 0.451 | 0.003 | <0.001 | 355 |
| Memory problems | | 2.56 (1.03) | 2.58 (1.01) | 2.62 (0.92) | 0.931 | <0.001 | | | | 355 |
| Trauma at childhood | | 1.06 (1.17) | 0.96 (1.17) | 1.73 (1.42) | 0.001 | 0.0221 | 0.583 | 0.002 | 0.004 | 355 |

*(Continued)*

**Table 2.** (Continued)

Survey of autonomic symptoms

| Score | Control (n=73) | Long COVID (n=143) | ME/CFS (n=170) | hEDS (n=290) | P value | | | | | Missing[c] |
|---|---|---|---|---|---|---|---|---|---|---|
| | | | | | Over-all[a] | ε² | ME/CFS-Long COVID [b] | ME/CFS-hEDS[b] | Long COVID -hEDS[b] | |
| Pelvic pain | | 0.95 (1.16) | 1.08 (1.19) | 1.89 (1.28) | <0.001 | 0.0412 | 0.423 | <0.001 | <0.001 | 355 |

Data are mean±sd. %, Prevalence of respective variable in percent. ε², Epsilon squared. [a] = Calculated using Kruskal-Wallis or chi-squared test as appropriate. [b] = Pairwise comparison calculated using Dunn or Fisher Exact test adjusted by Holm method as appropriate. [c] = missing values in the dataset are primarily due to the Central Sensitization Inventory (CSI) not being administered to control participants and to early hypermobile Ehlers-Danlos syndrome (hEDS) patients.

of central sensitization such as chronic pain, brain fog, fatigue and autonomic complaints has been documented in a variety of pain and fatigue-related syndromes, including [55], long COVID [58], hEDS [59], and chronic fatigue syndrome [56]. Our study confirmed a high prevalence of central sensitization in Long COVID (78.1%), ME/CFS (85.4%), and hEDS (92.4%). The high prevalence of central sensitization in these conditions likely contributes to the significant symptoms burden experienced by patients.

## Cerebral blood flow

In both Long COVID and ME/CFS, orthostatic CBFv was reduced, due either to abnormal cerebral autoregulation (consistent with OCHOS) or due to hypocapnia-induced cerebral arteriolar vasoconstriction (consistent with POTS and HYCH). Long COVID patients had more frequent orthostatic cerebral blood flow abnormalities and a greater decline in orthostatic cerebral blood flow than ME/CFS patients. Orthostatic hypotension did not play a significant role, because it was only detected in a few patients and orthostatic blood pressure remained in an autoregulatory range.

Reduced CBFv and associated cerebral hypoperfusion may explain some of the disabling symptoms of Long COVID and ME/CFS, such as lightheadedness, brain fog, and chronic fatigue. Previous studies have shown correlations between declines in orthostatic CBFv and lightheadedness [60], a key symptom of cerebral hypoperfusion. Typically a reduction in orthostatic CBFv by 19% or more from the supine baseline is associated with symptoms of central nervous system dysfunction [61,62]. Both our patient groups exceeded that level of decline (Long COVID −25% and ME/CFS −22%). In addition to the reduction of cerebral blood flow, respiratory alkalosis associated with hypocapnia changes neuronal excitability and may alter brain activity [63,64]. Imaging studies using arterial spin labeling showed reduced cerebral blood flow in COVID-19 patients [65,66]. Cerebral hypoperfusion consistent with a large resting state central network dysfunction was detected in Long COVID [67]. Cerebral blood flow dysregulation is also present in hEDS although the abnormality is less severe compared to Long COVID and ME/CFS patients.

## Autonomic features

Our study detected frequent autonomic failure in both Long COVID (95%) and ME/CFS (89%). Autonomic failure was widespread, affecting cardiovagal, adrenergic, and sudomotor domains. While the cardiovagal and adrenergic abnormalities were mild, sudomotor abnormalities were moderate. Although the dysautonomia tended to be worse in Long COVID compared to ME/CFS, the pattern was similar, affecting multiple domains simultaneously.

Orthostatic intolerance associated with autonomic dysregulation has been observed in both Long COVID and ME/CFS, although the reports are inconsistent [15]. In ME/CFS, most spectral analysis studies showed decreased heart rate variability with reduced parasympathetic and sympathetic activity, with increasing the sympathetic/parasympathetic ratio [13]

**Table 3. Results of autonomic testing.**

| Variable | Control (n = 73) | Long COVID (n = 143) | ME/CFS (n = 170) | hEDS (n = 290) | P value | | | | | Missing |
|---|---|---|---|---|---|---|---|---|---|---|
| | | | | | Over-all[a] | ε² | ME/CFS-Long COVID [b] | ME/CFS-hEDS[b] | Long COVID-hEDS[b] | |
| Deep breathing, heart rate, beats/minute | 16.20 (8.09) | 13.59 (8.04) | 12.68 (7.14) | 14.45 (7.65) | 0.002 | 0.022 | 0.487 | 0.019 | 0.111 | 0 |
| Valsalva ratio, beats/minute | 1.61 (0.25) | 1.74 (1.79) | 1.54 (0.31) | 2.17 (7.98) | 0.027 | 0.014 | 0.564 | 0.052 | 0.184 | 1 |
| Valsalva maneuver, end of phase 2 decline, mmHg | 8.32 (9.42) | −5.87 (12.89) | −6.39 (16.14) | −6.10 (14.85) | <0.001 | 0.104 | 0.564 | 0.052 | 0.184 | 0 |
| Electrochemical skin conductance, uS | 82.96 (7.79) | 77.70 (15.21) | 75.76 (14.94) | 80.07 (11.98) | 0.004 | 0.02 | 0.138 | 0.003 | 0.299 | 79 |
| Electrochemical skin conductance, uS/kg | 1.41 (0.21) | 1.03 (0.32) | 1.11 (0.34) | 1.21 (0.33) | <0.001 | 0.053 | 0.032 | 0.008 | <0.001 | 79 |
| Epidermal nerve fiber density at proximal thigh, fibers/mm | 13.44 (3.57) | 11.86 (4.55) | 12.78 (4.61) | 12.07 (4.53) | 0.033 | 0.013 | 0.208 | 0.239 | 0.859 | 7 |
| Epidermal nerve fiber density at calf, fibers/mm | 10.15 (2.25) | 7.94 (3.46) | 8.79 (3.55) | 8.20 (4.00) | <0.001 | 0.051 | 0.043 | 0.049 | 0.622 | 0 |
| Sweat gland nerve fiber density at proximal thigh, % of grid | 57.72 (9.98) | 55.87 (14.84) | 59.27 (17.44) | 52.75 (16.63) | 0.009 | 0.017 | 0.043 | 0.049 | 0.622 | 293 |
| Sweat gland nerve fiber density at calf, % of grid | 49.08 (10.59) | 49.09 (17.12) | 49.00 (17.34) | 45.91 (19.54) | 0.381 | 0.005 | | | | 176 |
| Heart rate supine, beats per minute | 74.11 (12.55) | 73.75 (12.18) | 72.28 (11.96) | 77.34 (13.70) | <0.001 | 0.025 | 0.438 | <0.001 | 0.028 | 0 |
| Heart rate orthostatic, beats per minute | 88.71 (14.49) | 92.06 (19.56) | 89.75 (18.17) | 97.51 (18.96) | <0.001 | 0.037 | 0.514 | <0.001 | 0.005 | 0 |
| Systolic BP supine, mmHg | 116.85 (8.91) | 124.55 (14.21) | 122.31 (17.09) | 116.11 (12.83) | <0.001 | 0.064 | 0.035 | <0.001 | <0.001 | 0 |
| Systolic blood pressure orthostatic, mmHg | 114.07 (9.86) | 118.78 (15.48) | 119.43 (18.38) | 114.68 (13.56) | 0.001 | 0.012 | 0.828 | 0.114 | 0.166 | 0 |
| Mean blood pressure supine, mmHg | 88.60 (7.11) | 93.32 (9.08) | 91.58 (10.80) | 88.45 (9.49) | <0.001 | 0.048 | 0.047 | 0.003 | <0.001 | 0 |
| Mean blood pressure orthostatic, mmHg | 88.17 (7.25) | 92.76 (10.47) | 92.35 (11.78) | 90.73 (10.60) | 0.009 | 0.015 | 0.63 | 0.355 | 0.191 | 0 |
| Diastolic blood pressure, mmHg, supine | 74.48 (6.84) | 77.70 (7.95) | 76.21 (8.72) | 74.61 (8.53) | 0.002 | 0.031 | 0.08 | 0.042 | <0.001 | 0 |
| Diastolic blood pressure, mmHg, orthostatic | 75.22 (6.80) | 79.76 (8.79) | 78.81 (9.51) | 78.75 (9.78) | 0.007 | 0.019 | 0.525 | 0.963 | 0.62 | 0 |
| Systolic CBFv supine, cm/sec | 108.48 (10.96) | 92.20 (18.68) | 93.30 (18.41) | 100.96 (17.53) | <0.001 | 0.095 | 0.697 | <0.001 | <0.001 | 0 |
| Systolic CBFv orthostatic, cm/sec | 99.99 (11.03) | 76.59 (18.46) | 80.48 (16.36) | 86.45 (17.93) | <0.001 | 0.147 | 0.116 | <0.001 | <0.001 | 0 |
| Mean CBFv supine, cm/sec | 67.47 (7.36) | 58.88 (12.64) | 59.13 (12.77) | 64.62 (12.06) | <0.001 | 0.07 | 0.868 | <0.001 | <0.001 | 0 |
| Mean CBFv orthostatic, cm/sec | 63.36 (7.65) | 48.44 (11.85) | 50.52 (10.99) | 55.41 (12.67) | <0.001 | 0.143 | 0.198 | <0.001 | <0.001 | 0 |
| Diastolic CBFv supine, cm/sec | 46.96 (6.96) | 42.34 (10.29) | 42.07 (10.69) | 46.49 (10.21) | <0.001 | 0.042 | 0.198 | <0.001 | <0.001 | 0 |
| Diastolic CBFv orthostatic, cm/sec | 45.04 (7.29) | 34.37 (9.30) | 35.55 (9.33) | 39.89 (11.23) | <0.001 | 0.113 | 0.364 | <0.001 | <0.001 | 0 |
| Mean CBFv corrected for $CO_2$ orthostatic, cm/sec | 69.47 (8.17) | 54.97 (12.76) | 55.31 (12.97) | 62.41 (14.32) | <0.001 | 0.136 | 0.364 | <0.001 | <0.001 | 0 |
| Maximal decline in orthostatic mean CBFv, cm/sec | −5.66 (2.38) | −15.07 (8.19) | −13.52 (7.55) | −14.44 (7.86) | <0.001 | 0.161 | 0.24 | 0.327 | 0.601 | 0 |

*(Continued)*

**Table 3.** (Continued)

| Variable | Control (n = 73) | Long COVID (n = 143) | ME/CFS (n = 170) | hEDS (n = 290) | Over-all[a] | ε² | ME/CFS-Long COVID [b] | ME/CFS-hEDS[b] | Long COVID -hEDS[b] | Missing |
|---|---|---|---|---|---|---|---|---|---|---|
| Maximal decline in orthostatic mean CBFv, % | −8.35 (3.18) | −24.97 (11.05) | −22.16 (10.46) | −22.11 (11.19) | <0.001 | <0.001 | 0.37 | 1 | 0.146 | 0 |
| Respiratory frequency supine, breaths per minute | 15.16 (3.94) | 15.64 (5.71) | 15.34 (5.29) | 15.48 (4.91) | 0.84 | 0.001 | | | | 42 |
| Respiratory frequency orthostatic, breaths per minute | 14.87 (2.36) | 16.61 (8.39) | 15.42 (5.64) | 15.67 (5.35) | 0.573 | 0.003 | | | | 42 |
| End-tidal $CO_2$ supine, mmHg | 37.34 (3.14) | 33.76 (4.11) | 34.94 (4.32) | 35.15 (3.92) | <0.001 | <0.001 | 0.004 | 0.956 | 0.001 | 0 |
| End-tidal $CO_2$ orthostatic, mmHg | 34.04 (3.05) | 28.86 (6.04) | 31.66 (5.01) | 30.61 (6.08) | <0.001 | 0.082 | <0.001 | 0.149 | 0.004 | 0 |
| Minimal end-tidal $CO_2$, mmHg, orthostatic | 33.33 (2.95) | 25.73 (6.06) | 28.15 (5.39) | 27.52 (6.35) | <0.001 | <0.001 | 0.002 | 0.431 | 0.004 | 0 |
| Maximal decline in orthostatic end-tidal $CO_2$, mmHg | −4.01 (0.91) | −8.03 (4.93) | −6.79 (4.62) | −7.63 (5.35) | <0.001 | 0.062 | 0.053 | 0.236 | 0.268 | 0 |
| Maximal decline in orthostatic end-tidal $CO_2$, % | −10.76 (2.26) | −23.95 (14.71) | −19.22 (12.98) | −21.85 (15.44) | <0.001 | 0.086 | 0.016 | 0.254 | 0.094 | 0 |
| CVRi supine, mmHg/cm/sec | 1.33 (0.19) | 1.68 (0.47) | 1.64 (0.51) | 1.42 (0.32) | <0.001 | <0.001 | 0.262 | <0.001 | <0.001 | 0 |
| CVRi orthostatic, mmHg/cm/sec | 1.09 (0.18) | 1.62 (0.59) | 1.51 (0.46) | 1.35 (0.40) | <0.001 | 0.134 | 0.104 | <0.001 | <0.001 | 0 |
| Cerebrovascular reactivity, %/mmHg | 1.48 (0.76) | 2.66 (2.46) | 2.45 (3.68) | 2.66 (2.88) | <0.001 | 0.029 | 0.862 | 0.898 | 0.749 | 0 |
| QASAT-CBFv, tilt response, range 0–10 | 0.00 (0.00) | 5.90 (3.36) | 5.27 (3.40) | 4.74 (3.68) | <0.001 | 0.228 | 0.139 | 0.141 | 0.003 | 0 |
| QASAT-ET-$CO_2$, tilt response, range 0–10 | 0.05 (0.47) | 3.05 (3.63) | 2.59 (3.45) | 3.04 (3.75) | <0.001 | 0.092 | 0.237 | 0.176 | 0.96 | 0 |
| QASAT-Autonomic failure, range 0–22 | 0.00 (0.00) | 4.55 (3.17) | 4.12 (3.22) | 3.41 (2.51) | <0.001 | 0.259 | 0.142 | 0.057 | <0.001 | 0 |
| QASAT-Cardiovagal, range 0–3 | 0.00 (0.00) | 0.43 (0.63) | 0.43 (0.69) | 0.45 (0.71) | <0.001 | 0.054 | 1 | 0.953 | 0.884 | 0 |
| QASAT-Adrenergic, range 0–3 | 0.00 (0.00) | 1.05 (0.84) | 1.11 (0.95) | 1.09 (0.97) | <0.001 | 0.136 | 1 | 0.82 | 0.983 | 0 |
| QASAT-Orthostatic hypotension, range 0–10 | 0.00 (0.00) | 0.73 (1.94) | 0.56 (1.62) | 0.42 (1.21) | <0.001 | 0.026 | 0.532 | 0.547 | 0.243 | 0 |
| QASAT-Orthostatic tachycardia, range 0–10 | 0.00 (0.00) | 1.55 (2.86) | 1.28 (2.73) | 1.86 (3.08) | <0.001 | <0.001 | 0.313 | 0.002 | 0.067 | 0 |
| QASAT-Sudomotor, range ESC 0–6 | 0.00 (0.00) | 2.37 (1.85) | 2.04 (1.82) | 1.45 (1.67) | <0.001 | 0.069 | 0.115 | <0.001 | <0.001 | 79 |
| QASAT-ENFD, range 0–8 | 0.00 (0.00) | 1.30 (1.83) | 0.84 (1.52) | 1.54 (2.03) | <0.001 | 0.105 | 0.011 | <0.001 | 0.263 | 0 |
| QASAT-SGNFD, range 0–8 | 0.00 (0.00) | 0.65 (1.39) | 0.68 (1.23) | 0.95 (1.74) | <0.001 | 0.044 | 0.761 | 0.403 | 0.331 | 104 |

Data are mean±sd. %, Prevalence of respective variable in percent. ε², Epsilon squared. [a] = calculated using Kruskal-Wallis test. [b] = pairwise comparison calculated using Dunn test.

CVRi= cerebrovascular resistance index; QASAT range denotes from 0 = normal, >0 =abnormal.

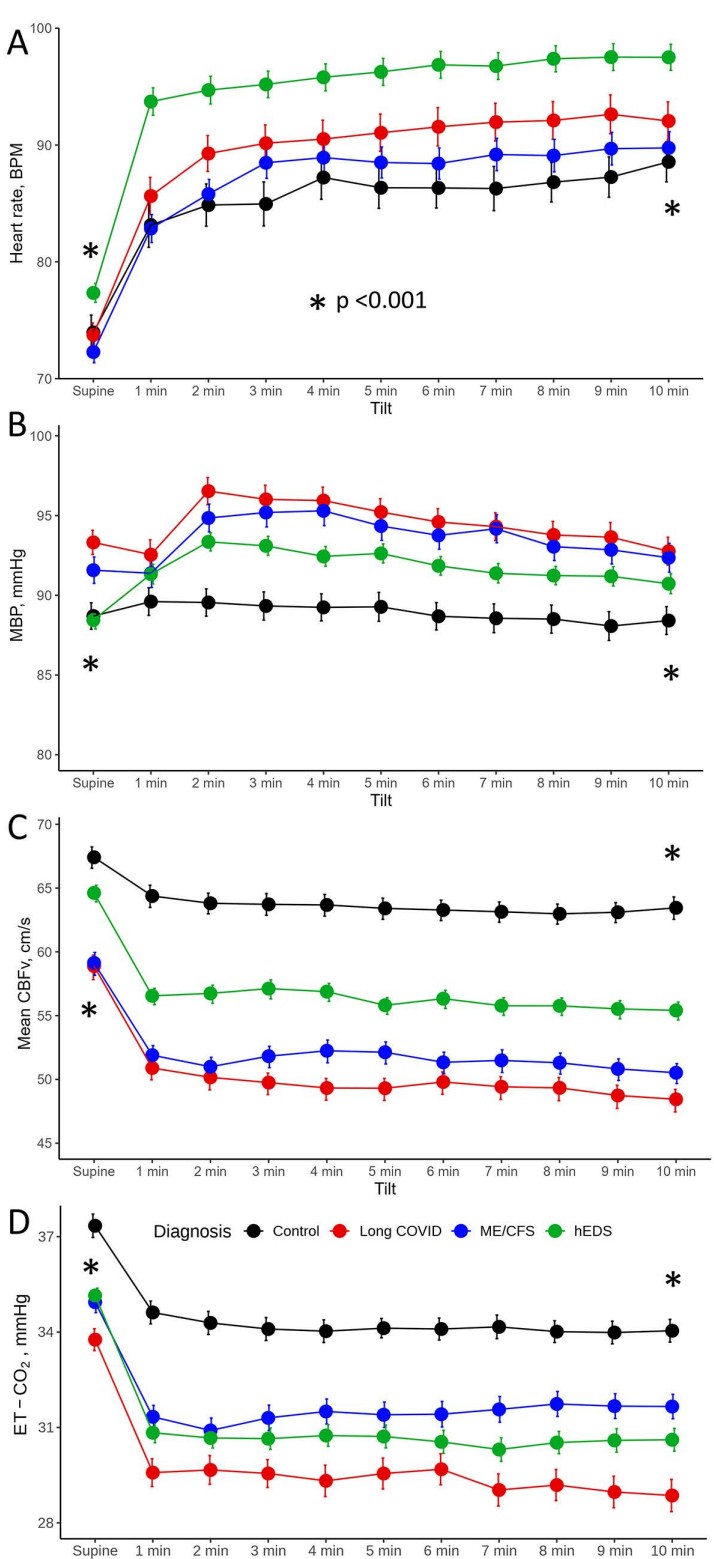

**Fig 2. The head-up tilt profile showing hemodynamic variables at supine baseline and at every minute of head-up tilt, expressed as mean±sd.**
A: heart rate; B: mean blood pressure; C: mean cerebral blood flow velocity in the middle cerebral artery (CBFv); D: end-tidal $CO_2$. *Denotes overall p value calculated by ANOVA.

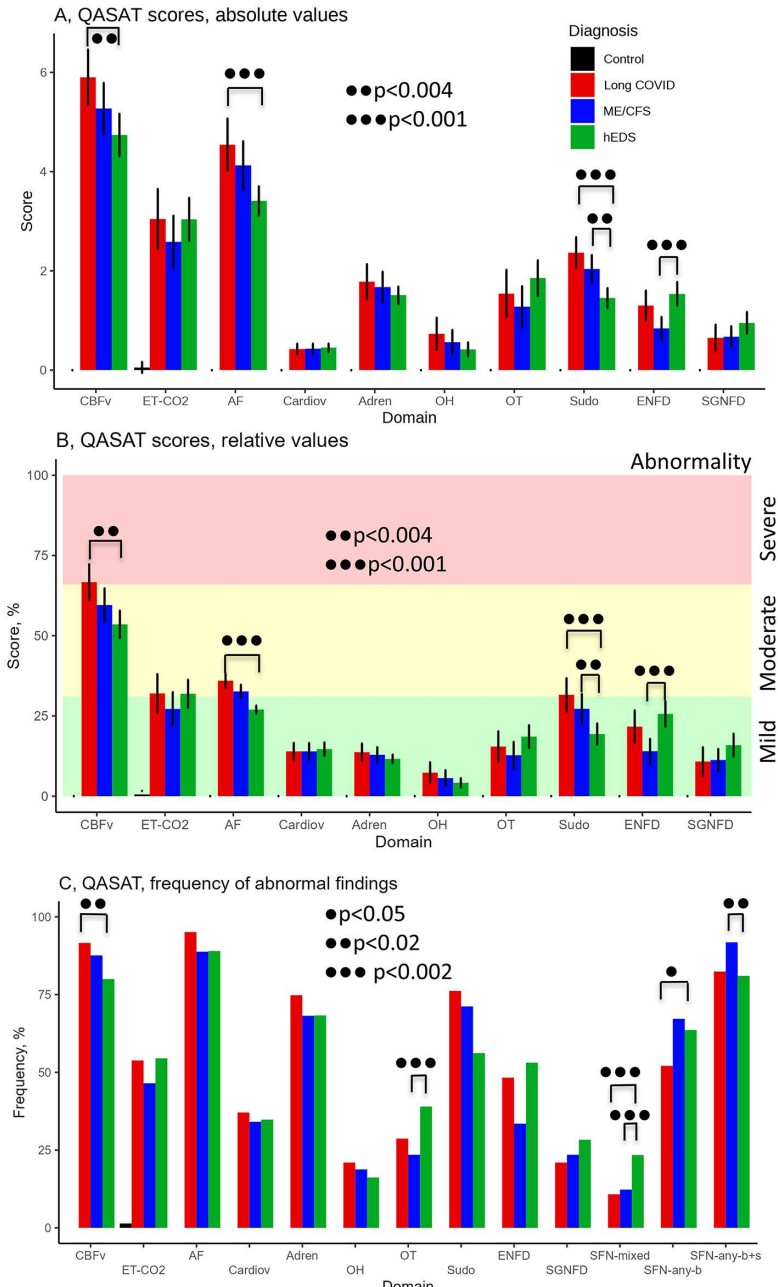

**Fig 3. QASAT results. (A)** Absolute scores, mean±sd. **(B)** Normalized scores in percent, mean±sd.int. **(C)** Percentage of patients in which the QASAT score was abnormal (> 0). CBFv = cerebral blood flow velocity, ET-CO$_2$ = end-tidal CO$_2$, AF = autonomic failure; OT = orthostatic tachycardia (orthostatic heart rate increment ≥ 30 BPM); Sudo = sudomotor; OH = orthostatic hypotension; SFN = small fiber neuropathy; ENFD = epidermal nerve fiber density, SGNFD = sweat gland nerve fiber density, SFN-any-b = small fiber neuropathy detected on skin biopsy defined as abnormal ENFD or SGNFD, SFN-any-b + s = small fiber neuropathy detected on skin biopsy or sudomotor testing.

that was interpreted as relative sympathetic overactivity, although that interpretation is debated [68]. Another study found baroreflex failure with vagal efferent defect derived from analysis of blood pressure variability [15]. Yet another study found no difference in objective autonomic testing compared to chronic fatigue and controls [69]. The variability in findings can

**Table 4. Frequency of abnormal findings.**

| Variable | Control (n = 73) | Long COVID (n = 143) | ME/CFS (n = 170) | hEDS (n = 290) | P value | | | | Missing |
|---|---|---|---|---|---|---|---|---|---|
| | | | | | Overall[a] | ME/CFS-Long COVID[b] | ME/CFS-hEDS[b] | Long COVID -hEDS[b] | |
| Orthostatic lightheadedness/dizziness,% | 8.0 | 65.7 | 65.9 | 72.4 | <0.001 | 0.999 | 0.426 | 0.426 | 0 |
| Orthostatic dyspnea,% | 0.0 | 37.1 | 28.2 | 21.0 | <0.001 | 0.177 | 0.177 | 0.001 | 0 |
| QASAT-CBFv, reduced during the tilt, % | 0.0 | 91.6 | 87.6 | 80.0 | <0.001 | 0.274 | 0.081 | 0.006 | 0 |
| QASAT-ET-$CO_2$, reduced during the tilt, % | 1.4 | 53.8 | 46.5 | 54.5 | <0.001 | 0.426 | 0.304 | 0.919 | 0 |
| QASAT-Autonomic failure, % | 0.0 | 95.1 | 88.8 | 89.0 | <0.001 | 0.144 | 0.999 | 0.144 | 0 |
| QASAT-Cardiovagal, % | 0.0 | 37.1 | 34.1 | 34.8 | <0.001 | 0.999 | 0.999 | 0.999 | 0 |
| QASAT-Adrenergic,% | 0.0 | 74.8 | 68.2 | 68.3 | <0.001 | 0.539 | 0.999 | 0.539 | 0 |
| QASAT-Orthostatic hypotension, % | 0.0 | 21.0 | 18.8 | 16.2 | <0.001 | 0.999 | 0.999 | 0.693 | 0 |
| QASAT-Orthostatic tachycardia,% | 0.0 | 28.7 | 23.5 | 39.0 | <0.001 | 0.304 | 0.002 | 0.085 | 0 |
| QASAT-Sudomotor,% | 0.0 | 77.3 | 72.0 | 56.6 | <0.001 | 0.299 | 0.003 | <0.001 | 6 |
| QASAT-ENFD,% | 0.0 | 48.3 | 33.5 | 53.1 | <0.001 | 0.0216 | <0.001 | 0.359 | 0 |
| QASAT-SGNFD,% | 0.0 | 27.8 | 28.8 | 32.2 | <0.001 | 0.999 | 0.999 | 0.999 | 104 |
| SFN, mixed,% | 0.0 | 11.9 | 10.8 | 23.0 | <0.001 | 0.851 | 0.007 | 0.019 | 50 |
| SFN, any from biopsy,% | 0.0 | 67.2 | 52.6 | 63.3 | <0.001 | 0.0437 | 0.0787 | 0.499 | 54 |
| SFN, any,% | 0.0 | 91.4 | 82.9 | 81.1 | <0.001 | 0.0804 | 0.703 | 0.019 | 82 |
| Postural tachycardia syndrome (POTS),% | 0.0 | 22.4 | 19.4 | 32.4 | <0.001 | 0.577 | 0.008 | 0.066 | 0 |
| Hypocapnic cerebral hypoperfusion (HYCH),% | 0.0 | 23.8 | 21.8 | 21.0 | <0.001 | 0.999 | 0.999 | 0.999 | 0 |
| Orthostatic cerebral hypoperfusion syndrome (OCHOS),% | 0.0 | 25.9 | 31.8 | 17.6 | <0.001 | 0.264 | 0.002 | 0.113 | 0 |
| Neurogenic orthostatic hypotension,% | 0.0 | 14.7 | 14.7 | 9.7 | <0.001 | 0.999 | 0.389 | 0.389 | 0 |

Data are mean±sd. %=prevalence of abnormal findings in percent. [a]=Calculated using chi-squared test. [b]=Pairwise comparison calculated using Fisher Exact test.

be attributed to the heterogeneity of ME/CFS, along with differences in inclusion criteria and techniques for the evaluation. Mild autonomic abnormalities have been detected in Long COVID patients [3,70–73]. This current study confirmed our previous finding [3,73] and expanded the analysis to the ME/CFS group. Due to the small number of pre-Omicron cases, we did not perform a comparative analysis of the effects of different SARS-CoV-2 strains.

iCPET also showed similarities between Long COVID and ME/CFS including the prevalence of preload failure (96% vs. 92.4%) and deconditioning (64% vs. 64%). Cardiac output which is proportional to BMI, was lower in ME/CFS [74]. However, the Long COVID group had a higher BMI and cardiac output adjusted for BMI was similar between the groups. Although invasive iCPET was available only in a subset of patients, the similarities in key iCPET metrics support the notion of shared common pathophysiology in both conditions.

## Skin biopsies

Skin biopsies, which provide direct evidence of peripheral nerve damage, confirmed the presence of SFN in both Long COVID and ME/CFS patients. SFN was more frequently observed in individuals with Long COVID, although a similar prevalence was also noted in patients with hypermobile Ehlers-Danlos syndrome (hEDS).

**Table 5. Invasive Cardiopulmonary exercise test results.**

| Variable | Long COVID (n = 25) | ME/CFS (n = 66) | P-value[a] | Missing |
|---|---|---|---|---|
| Age, years | 46.52 (11.55) | 41.85 (14.06) | 0.088 | 0 |
| Gender, female % | 64 | 86 | 0.036 | 0 |
| Rest stroke volume, ml, | 94.12 (31.31) | 76.98 (21.03) | 0.013 | 0 |
| Exercise stroke volume, ml | 87.81 (21.79) | 74.73 (20.49) | 0.01 | 0 |
| Difference (exercise-rest) stroke volume, ml | −6.30 (25.76) | −2.26 (21.24) | 0.848 | 0 |
| Supine heart rate from autonomic testing, bpm | 69.76 (12.22) | 72.12 (12.83) | 0.426 | 0 |
| Rest heart rate, bpm | 77.12 (10.65) | 79.80 (13.66) | 0.455 | 0 |
| Exercise heart rate, bpm | 148.96 (25.04) | 140.88 (28.21) | 0.175 | 0 |
| BMI, m/kg$^2$ | 29.16 (4.83) | 26.06 (6.23) | 0.006 | 0 |
| Rest cardiac output, l/min | 7.11 (1.97) | 6.04 (1.49) | 0.018 | 0 |
| Exercise cardiac output, l/min | 13.02 (3.84) | 10.55 (3.64) | 0.003 | 0 |
| Rest cardiac output adjusted for BMI, l/min/m/kg2 | 0.25 (0.08) | 0.24 (0.08) | 0.752 | 0 |
| Exercise cardiac output adjusted for BMI, l/min/m/kg2 | 0.46 (0.17) | 0.42 (0.17) | 0.381 | 0 |
| Rest VO$_2$, ml/min | 363.24 (126.94) | 293.52 (65.83) | 0.001 | 0 |
| Exercise VO$_2$, ml/min | 1669.40 (693.50) | 1153.55 (559.00) | <0.001 | 0 |
| Rest right atrial pressure, mmHg | 0.16 (1.14) | −0.14 (1.98) | 0.494 | 0 |
| Exercise right atrial pressure, mmHg | 1.52 (2.82) | 1.50 (3.13) | 0.631 | 0 |
| Preload failure, % | 96.0 | 92.4 | 0.542 | 0 |
| Peak VO$_2$, % predicted | 82.80 (17.66) | 78.68 (23.53) | 0.21 | 0 |
| Deconditioning, % | 64.0 | 63.6 | 0.974 | 0 |
| Peak cardiac output, % predicted | 94.49 (17.34) | 96.59 (25.78) | 0.953 | 0 |
| Anaerobic threshold, % predicted | 46.47 (15.52) | 46.44 (13.33) | 0.528 | 0 |
| Peripheral oxygen extraction | 0.88 (0.13) | 0.83 (0.12) | 0.092 | 0 |
| Mitochondrial myopathy, % | 20.0 | 41.5 | 0.096 | 0 |

VO$_2$, oxygen uptake. Deconditioning was defined as the predicted peak oxygen uptake < 85%, preload failure was defined as right atrial pressure < 6.5 mmHg, mitochondrial myopathy was defined as $(CaO_2 − CvO_2)/Hb < 0.8$, where $CaO_2$ = arterial oxygen content, $VaO_2$ = venous oxygen content, and Hb = hemoglobin. [a] = Calculated using Kruskal-Wallis test or chi-squared test as appropriate.

### Inflammatory, metabolic, and hormonal markers

We were unable to find particular features in laboratory values that would differentiate the studied disorders. Most of the subjects had normal laboratory values, and abnormal results were found in a minority of patients. There were no differences between Long COVID and ME/CFS in inflammatory, autoimmune, adrenergic, and hormonal markers. These findings are consistent with the hypothesis that both disorders may share a common pathophysiological mechanism. However, the failure to detect elevated inflammatory or autoimmune markers does not support an inflammatory or autoimmune theory for either disorder. Neverthelles, the tests used in our study may not be sensitive enough to detect low-grade inflammation. Elevated cytokines, including IL1β, IL6, and TNFα, have been reported in some, but not all, studies of Long COVID [75]. No differences were found in our study.

Furthermore, we were unable to document differences in levels of norepinephrine, a marker of adrenergic functions, which can be abnormal in peripheral dysautonomia [76]. Nevertheless, norepinephrine levels in peripheral blood do not correlate with central adrenergic activity [77]; therefore, our study cannot rule out central adrenergic dysregulation. Although hormonal dysregulation has been implicated in Long COVID [78], our study did not confirm this finding. We were unable to detect hormonal changes indicative of adrenal or hypothalamic-pituitary-adrenal axis insufficiency, as evidenced by normal cortisol and ACTH levels across our studied groups [79].

**Table 6. Results summary showing differences Long COVID vs. ME/CFS.**

| Variable | Long COVID (n=143) | ME/CFS (n=170) | Difference | P Value |
|---|---|---|---|---|
| Age, years | 43.71 (13.23) | 44.45 (14.69) | −0.74 | 0.880 |
| Gender, female % | 72.7 | 78.8 | −6.1 | 0.233 |
| **Patient's reported outcome measures** | | | | |
| Survey of autonomic symptoms | 22.50 (10.02) | 23.42 (9.17) | −0.92 | 0.879 |
| Neuropathy total symptom score-6 | 9.59 (4.83) | 9.65 (5.13) | −0.06 | 0.771 |
| Numerical rating pain scale | 2.71 (2.84) | 2.90 (2.69) | −0.19 | 0.492 |
| Central sensitization syndrome, % | 78.1 | 85.4 | −7.3 | 0.410 |
| **Autonomic testing** | | | | |
| Maximal decline in orthostatic mean CBFv, % | −24.97 (11.05) | −22.16 (10.46) | 2.81 | 0.370 |
| Maximal decline in orthostatic end-tidal $CO_2$, % | −23.95 (14.71) | −19.22 (12.98) | 4.73 | 0.016 |
| QASAT-Autonomic failure, range 0–22 | 4.55 (3.17) | 4.12 (3.22) | 0.43 | 0.142 |
| SFN, any from biopsy, % | 67.2 | 52.6 | 14.6 | 0.044 |
| Postural tachycardia syndrome (POTS), % | 22.4 | 19.4 | 3.0 | 0.577 |
| Hypocapnic cerebral hypoperfusion (HYCH), % | 23.8 | 21.8 | 2.0 | 0.999 |
| Orthostatic cerebral hypoperfusion syndrome (OCHOS),% | 25.9 | 31.8 | −5.9 | 0.264 |
| Neurogenic orthostatic hypotension, % | 14.7 | 14.7 | 0.0 | 0.999 |
| **Invasive cardiopulmonary exercise** | | | | |
| | (n=25) | (n=66) | | |
| Preload failure, % | 96.0 | 92.4 | 3.6 | 0.542 |
| Deconditioning, % | 64.0 | 63.6 | 0.4 | 0.974 |

Summary of the main results from Tables 1–5, and differences of the means. P values indicate pairwise comparisons calculated using Fisher exact test or Kruskal-Wallis test.

## Comparisons of Long COVID/ME/CFS to hEDS

While laboratory blood evaluations were unable to distinguish among the three disorders, surveys, and autonomic functional assessments with skin biopsies revealed differentiating features. hEDS subjects reported more severe sensory and autonomic symptoms compared to Long COVID and ME/CFS. Although some overlap was observed, hEDS was associated with less severe cerebrovascular dysregulation but more pronounced peripheral neurodegeneration, as evidenced by greater sudomotor dysfunction and more frequent and severe small fiber neuropathy.

## Study limitations

We are a referral center for dysautonomia, so we may not see a representative group of patients. We also used historical controls. However, the large number of patients we have studied may still provide a representative sample of these patient populations.

A limitation of direct Long COVID and ME/CFS comparison is the fact that the duration of the symptoms was much longer in ME/CFS. Duration of the disease may affect the signature of ME/CFS, particularly the immunological profile [80] which was not different between the groups. A greater prevalence of deconditioning, which would be expected with a longer-lasting ME/CFS, was also not detected in our study. Furthermore, autonomic failure was more severe in the Long COVID group, also speaking against the time effect. Nevertheless, it would be useful to longitudinally observe ME/CFS and Long COVID to determine whether these entities converge into an indistinguishable syndrome, which would provide additional evidence about the common pathophysiology of both disorders.

The lack of laboratory values in healthy controls is another study limitation. However, all laboratory tests were validated in a clinical setting, have established normative data and performed at CLIA-certified laboratories.

Methodologically, cerebral blood flow was assessed indirectly using transcranial Doppler, which measures flow velocity, and not flow directly. The velocity is proportional to blood flow, assuming that the diameter of the insonated vessel does not change during orthostatic stress, which was confirmed by an imaging study [81]. CBFv is also affected by the angle of the transcranial Doppler probe. Although the angle varies from patient to patient, once the probe was properly positioned and stabilized with a 3D holder, the same angle was maintained throughout the testing.

## Conclusion

We found evidence of similar prevalence of central sensitization and similar patterns of dysregulation in cerebrovascular blood flow, respiratory and cardiovascular autonomic reflexes, and small fiber neuropathy in both Long COVID and ME/CFS. Hence, the large proportion of patients with these disorders likely lies along a spectrum with similar pathophysiology, at least as far as it concerns the cerebrovascular and autonomic nervous system, and, in principle, might benefit from similar therapeutic interventions. The ability to quantify cerebrovascular and autonomic dysfunction is helpful and it can provide the metric for therapeutic interventions. However, key findings (cerebrovascular, respiratory and cardiovascular dysregulation along with neurodegeneration) are not necessarily exclusive to Long COVID and ME/CFS since similar findings but with different distributions were found in hEDS, a condition with different cause. Further research should clarify whether these conditions share a common pathophysiological pathway or represent distinct but overlapping syndromes.

## Acknowledgments

The authors thank Diana Arevalo for helping with data collection.

## Author contributions

**Conceptualization:** Peter Novak.

**Data curation:** Peter Novak, David M. Systrom, Alexandra Witte, Sadie P. Marciano, Donna Felsenstein, Jeff M. Milunsky, Joel Krier.

**Formal analysis:** Peter Novak.

**Funding acquisition:** Peter Novak.

**Investigation:** Peter Novak.

**Methodology:** Peter Novak, David M. Systrom.

**Project administration:** Peter Novak.

**Resources:** Peter Novak.

**Software:** Peter Novak.

**Supervision:** Peter Novak, Mark C. Fishman.

**Writing – review & editing:** Alexandra Witte, Sadie P. Marciano, Donna Felsenstein, Jeff M. Milunsky, Aubrey Milunsky, Joel Krier, Mark C. Fishman.

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
