## [Decision Letter · Decision Letter 0]

1 Jul 2025

Dear Dr. Novak,

We look forward to receiving your revised manuscript.

Kind regards,

Hong-Liang Zhang, M.D., Ph.D.

Academic Editor

PLOS ONE

Journal Requirements:

2. Please expand the acronym “FBRI” (as indicated in your financial disclosure) so that it states the name of your funders in full.

This work was funded by Mona Taliaferro/Bay Shore Recycling, The National Heart, Lung and Blood Institute (NHLNI - 1OT2HL156812-01) and FBRI (2022A018462) to P. Novak.

4. In the online submission form, you indicated that restricted data will be available from the corresponding author upon reasonable request.

5. Please amend your list of authors on the manuscript to ensure that each author is linked to an affiliation. Authors’ affiliations should reflect the institution where the work was done (if authors moved subsequently, you can also list the new affiliation stating “current affiliation:….” as necessary).

6. Please include your tables as part of your main manuscript and remove the individual files. Please note that supplementary tables (should remain/ be uploaded) as separate "supporting information" files

8. Please remove all personal information, ensure that the data shared are in accordance with participant consent, and re-upload a fully anonymized data set.

Additional guidance on preparing raw data for publication can be found in our Data Policy (https://journals.plos.org/plosone/s/data-availability#loc-human-research-participant-data-and-other-sensitive-data) and in the following article: http://www.bmj.com/content/340/bmj.c181.long .

Reviewers' comments:

Reviewer's Responses to Questions

**Comments to the Author**

1. Is the manuscript technically sound, and do the data support the conclusions?

Reviewer #1: No

Reviewer #2: Yes

2. Has the statistical analysis been performed appropriately and rigorously?

Reviewer #1: No

Reviewer #2: Yes

3. Have the authors made all data underlying the findings in their manuscript fully available?

Reviewer #1: No

Reviewer #2: Yes

4. Is the manuscript presented in an intelligible fashion and written in standard English?

Reviewer #1: No

Reviewer #2: Yes

Reviewer #1: Your manuscript requires major revisions to improve clarity, structure, and the robustness of your analysis. The introduction lacks sufficient background information and references, making it difficult for readers unfamiliar with the topic to follow. The methods section needs better organization, clearer inclusion/exclusion criteria, and a more readable presentation of diagnostic tools like QASAT. The statistical analysis is incomplete—key details such as power calculations, handling of outliers, and confounder adjustments are missing, undermining confidence in the results. The results section is disorganized, presenting random patient factors without context, making interpretation difficult. The discussion lacks meaningful analysis of the data, failing to draw strong conclusions or explore clinical implications. Key recommendations: improve readability, ensure statistical rigor, clearly present patient characteristics, provide a structured interpretation of findings, and avoid excessive acronyms.

Reviewer #2: This manuscript studied the shared autonomic phenotype of Long COVID and ME/CFS, which is very interesting and important in post COVID era. The manuscript is beautifully written, and I truly enjoyed reading through the paper, in which the main conclusions are very well supported by the data presented. The authors have done an excellent job. Meaning while, I have few minor comments:

1. Figure resolution need be improved.

2.Data point should be showed in the figures, box plots and violin plots are preferred.

3. For long COVID patients, did the author considered the infection times? This should be discussed.

4. Why the authors choose ME/CFS for the comparation? More detailed explanation in the introduction section would be helpful.

**Do you want your identity to be public for this peer review?** For information about this choice, including consent withdrawal, please see our Privacy Policy

Reviewer #1: No

Reviewer #2: **Yes:** Shuo Yang

---

## [Author Response · Author response to Decision Letter 1]

7 Sep 2025

Dear Editor

We would like to express our sincere gratitude to both reviewers for their constructive feedback. We have carefully revised the paper in accordance with the suggestions. Below, we provide detailed responses to each of the reviewers' comments:

For authors

Peter Novak

Recommendation - Major Revision

Introduction

1. Recommend in the 1st few lines explaining how they “disable several organ systems” or give a one line explanation of the common symptoms and disabilities patients experience

Answer: Common symptoms associated with both conditions have been added to the introduction

2. Highlight what similarities are seen early in the introduction. A large portion of the beginning keeps referring to symptoms without enough background information

Answer: Most of the symptoms are shared in both conditions, that was now clarified in the introduction.

3. Missing references for the line starting with “ Both disorders.”, “They are speculated” and “Both disorders are believed to be…”

Answer: We added missing references

4. Missing references for “About 25%...”

Answer: We added missing references,

5. The line “Different elements..” needs to be rephrased - Remains too lengthy with multiple “and” conjunction in sentence structure

Answer: The sentence was rewritten

6. Recommend adding why hEDS patients were chosen as part of the control group or at least the clinical reasoning behind this addition of patients.

Answer: hEDS is a heritable disorder associated with dysautonomia.

Therefore, we aimed to assess whether the observed findings represent a common physiological response to illness or if they are specific to Long COVID and ME/CFS. This was added to the introduction.

7. Recommend adding what clinical question and hypothesis was aimed to be ascertained in the study. Needs more clarification

Answer: The tested hypothesis was added to the introduction.

Impression - Needs further background information of what was done in prior relationships and what other researchers had discovered about this phenom. I would also recommend more background information of why the comprehensive testing was selected in this way and explain deeper what the clinical question and purpose of the study. The introduction is written in a way whereby the onus remains on the reader to have a reasonable understanding of this topic to fully grasp the information which reduces the impact of the information following.

Answer: The introduction was expanded as outlined above.;

Materials and Methods

1. Recommend rephrasing the line starting with “ Hypocapnic cerebral hypoperfusion..”.

Answer: The sentence was revised.

2. Recommend rephrasing “ Orthostatic cerebral hypoperfusion…”. This line also has a closing bracket incorrectly

Answer: The closing bracket was corrected

3. Why was inclusion and exclusion criteria not bolded with the rest of the paragraph separators

Answer: The wrong bolding was corrected.

4. There was no statement about what age group was used in the inclusion/exclusion

Answer: The inclusion criterion includes the age 18 or older. This was inserted in the methods.

5. Recommend removing the “We” from “We followed the…”

Answer: The Sentence was revised.

6. Adding the criterias for diagnosis would aid the reader to understand how the patients were decided upon

Answer: The diagnostic criteria for the studied diagnoses was expanded.

7. How were healthy controls chosen before testing them?

Answer:

8. Recommend re - doing the QASAT paragraph into a more reading friendly manner. ?

Answer: The paragraph was rewritten to make it easier to read. The ranges were move to the tables.

Impression - Gaps noted in necessary parts of the materials and methods. Difficult to read and ascertain the important information. Recommendation to reduce the verbosity in certain key areas, summarize, rephrase and present the information in a simpler way to communicate the definitions and especially the QASAT paragraph better.

Statistical Analysis

1. Missing power analysis. What calculations were done to ascertain the study population needed for significant values

Answer: Power analysis was added to the methods.

2. What variables were continuous or categorical. Were values needed to be reorganized in different categories?

Answer: Most of variables were continuous. Variables assigned as % and without standard deviation in the bracket (for example percentage of abnormal findings) were categorical.

3. How was the data cleaned or pre processed? What was done for outliers

Answer: We did not remove outliers during data cleaning, as they often reflect clinically significant abnormalities rather than errors. From a statistical perspective, these values may appear as outliers; however, in a clinical context, they frequently represent real and meaningful abnormal results. Therefore, we retained these values to preserve the integrity of the clinical information. This decision is further discussed in the Methods section.

4. The study does not specify whether normality and homogeneity of variance assumptions were checked before performing ANOVA

Answer: For the continuous variables, we assessed the assumptions of normality and homogeneity of variance. Many of the variables showed non-normal distributions, and there was often a lack of homogeneity of variance between groups. Therefore, instead of using ANOVA, we recalculated data using the Kruskal-Wallis test, which is a non-parametric alternative that does not assume normal distribution and is robust to heterogeneity of variances across groups.

5. No reported random effects structure

Answer: This study did not include a random effects structure, as the design did not involve repeated measures or hierarchical/nested data that would require modeling random variability across subjects or groups. All analyses were conducted under the assumption of independent observations. If future studies involve multi-level or longitudinal data, incorporating random effects would be appropriate

6. No discussion of interaction effects (Group × Position). This would clarify whether different patient groups had distinct responses to tilt.

Answer: We added the results of the linear model to the results section and also we are discussing the overall difference between groups in the discussion.

7. What adjustments were made for possible confounders?

Answer: Yes, the calculations using linear models were adjusted for age and gender as confounders. We expanded the description of how the linear model was used.

8. Missing sensitivity analysis to assess impact of missing and ignoring data

Answer: The proportion of missing data was relatively low for both the survey data and autonomic testing, and was therefore deemed unlikely to bias the main findings. However, there were substantial missing values in the laboratory blood work, likely due to test ordering being at the discretion of the attending physicians.

To assess the potential impact of missing data on the results, we conducted a sensitivity analysis under the assumption that the data were missing at random (MAR). Specifically, we compared the results of the Kruskal–Wallis test for continuous variables across three scenarios: 11 Complete-case analysis (excluding missing values), 2. Mean imputation (replacing missing values with the overall mean), and 3. Median imputation (replacing missing values with the overall median).

The findings remained consistent across all three methods, suggesting that the missing values did not significantly affect the robustness of our results. Details of this analysis have been added to the Methods section.

The sensitivity analysis showed that missing values did not significantly affect the robustness of our results.

Impression - There is a surprising lack of crucial data to understand if the figures about to be presented in the result section are believable given the large gaps in the analysis. At this point the results can be heavily scrutinized and would completely undermine the study as the readers do not have enough structure to accept the figures.

While the basic analysis was explained, the details of the analysis and imputation methods for robustness checks of the data itself remain crucial to accept the results with expected gravitas. This would remain the most important of the study and heavily weighs on the Major revision recommendation

Answer: The details of statistical analysis have been added as outline above.

Results

1. This is the 1st time the study mentions how many patients were studied

Answer: We added a flow diagram to clarify the number of studied patients.

2. Why was the differentiation of long covid patients based on virus type only mentioned now? Recommend mentioning before in the Materials and methods as well.

Answer: We mention the virus strains (pre-Delta, Delta and Omicron) in the method section.

3. How was the differentiation of the viruses important then? This also was not mentioned in the statistical analysis

Answer: We had only a small number of patients from the pre-Delta (n=6) and Delta (n=10) periods, as noted in the Results section. Therefore, comparisons between these viral strains are not meaningful due to the limited sample size.

4. What symptoms were shorter duration?

Answer: Regarding symptom duration, we asked patients to report the length of time since disease onset or since symptoms first appeared. However, we did not specifically categorize individual symptoms by duration in this study.

To clarify this, we added “…symptom duration defined as the length of time since disease onset.” To the results section and to the table 1.

5. None of the 2nd paragraph makes sense. Why does it matter if the hEDS had more irritable bowel syndrome or mast cell activation syndrome? How does all of this fit in? Answer: Ehlers-Danlos syndrome (EDS) is associated with a variety of allergic complications, including mast cell activation, as well as multiple pain syndromes and gastrointestinal problems such as irritable bowel syndrome (IBS).

Hakim A, De Wandele I, O’Callaghan C, Pocinki A, Rowe P. Chronic fatigue in Ehlers-Danlos syndrome-Hypermobile type. Am J Med Genet C Semin Med Genet. 2017;175: 175–180. doi:10.1002/ajmg.c.31542

Gensemer C, Burks R, Kautz S, Judge DP, Lavallee M, Norris RA. Hypermobile Ehlers-Danlos syndromes: Complex phenotypes, challenging diagnoses, and poorly understood causes. Dev Dyn Off Publ Am Assoc Anat. 2021;250: 318–344. doi:10.1002/dvdy.220

6. We were not presented on the characteristics of patients so mentioning BMI, Fibromyalgia etc appear at random.

Answer: BMI was included as a standard demographic variable. The diagnosis of fibromyalgia is commonly seen in various pain syndromes, including ME/CFS, autonomic dysfunction, and especially small fiber neuropathy. Some studies suggest that patients diagnosed with fibromyalgia may, in fact, have underlying small fiber neuropathy. Given that a significant portion of our study population was on pain medications, we chose to retain the fibromyalgia diagnosis, as it aligns with their clinical presentation and treatment patterns.

7. What does the use of pressor medications mean? Are we discussing midodrine? Fludrocortisone? This needs to be clarified.

Answers: The types of pressor medications were clarified in the Table 1 legend. These medications are proamatine, fludrocortisone, pyridostigmine, and droxidopa

8. Is there a need for a separate paragraph for iCPET as this was more historical data and a small portion of the patients had this data available?

Answer: Although the number of patients undergoing iCPET was small, the results are valuable because iCPET is an invasive test that provides direct measurements of stroke volume, cardiac filling pressures, and other hemodynamic parameters. These unique findings support and confirm our non-invasive results.

Impression - A Difficult read overall with lacking of placement of the data and tables within the body of the results rather than in the appendix. Confusing presentation of random patient factors and data as well in the beginning. Recommend redoing the result sections and include the

graphs rather than rely solely on paragraph presentation. I do believe there are too many acronyms to remember in this paper and recommend not relying on them to deliver the data, it becomes very burdensome to read.

No confidence intervals were shown in the data to determine how much variability was noticed.

Discussion

1.

2. Regarding reduced cerebral blood flow, Given the data or lack thereof, did patients with further reduction in CBFv have different or worse symptoms than others?

Did they experience more brain fog and was there a discussion to perform standardized cognitive tests to ascertain that relationship? This appears to be a big miss only to detect the reduction in flow and not have it impact for a clinical connection?

Answer: These are important and insightful questions. While we did not conduct standardized cognitive testing in this particular study, there is a substantial body of literature showing that cognitive function is impaired in both Long COVID and ME/CFS. We agree that establishing a direct clinical connection between reduced cerebral blood flow velocity (CBFv) and cognitive performance would significantly strengthen the findings. In fact, in a recent publication from our group, we demonstrated that reductions in cerebral blood flow correlated with symptom severity in the short term—suggesting that reduced CBF may indeed be related to cognitive decline. However, the absence of standardized cognitive assessments in this study is a limitation. Future research would benefit from incorporating such testing to directly correlate objective cognitive measures with cerebral blood flow metrics. This would help clarify the clinical implications of reduced CBFv, particularly regarding symptoms like brain fog and cognitive dysfunction.

2. A note for the longitudinal aspect of ME/CFS vs Long covid however was there a difference in patients by their strain? Was there a difference with patients who had long covid for a prolonged period than other patients?

Answer: That is interesting questions. Due to the small number of pre-Omicron cases, we did not perform a comparative analysis of the effects of different SARS-CoV-2 strains. This is mentioned in the discussion.

3. Including a limitation in the middle of the discussion is not warranted. Recommend removing that paragraph and summarizing in the end with the rest of the others.

Answer: The paragraph was moved to the limitation section.

4. What markers are noted? This was not mentioned before apart from CRP and Interleukin 6

Answer: The details if inflammatory markers were inserted in the methods.

5. How were the markers collected? Was this mentioned in the methods?

Answer: The markers were collected as a of routine clinical evaluations. This is mentioned in the methods.

6. Multiple missing references noted

Answer: The missing references were inserted.

7. Typographical error in limitations “a large number many patients”

Answer: The typo was corrected

Impression - Overall a disappointing discussion lacking actual true analysis and speculation on the results found. A noticeable part of it was discussing the studies against the findings for orthostatic hypotension and other parts not fully described e.g. the biopsies done. There was no acceptable differentiation of the different phenotypes of where they differed. It appears the data was collected, not properly presented or explained and just laid out instead of interpreting the data to explain what differences, similarities between the phenotypes and how it relates to a clinical picture. Much of the focus remains on trying to convince the reader the similarities without actually explaining the data. It was striking how little was relayed about the results of the patients and consistently comparing with the different groups especially the healthy control group as I believe they were almost never mentioned. Much more details and deliberation would be needed for this study to make clinical sense as currently it is poorly detailed statistical analysis to detect certain anomalies in these patients then trying to find similarities

---

## [Decision Letter · Decision Letter 1]

26 Oct 2025

Dear Dr. Novak,

Thank you for submitting your manuscript to PLOS ONE. After careful consideration, we feel that it has merit but does not fully meet PLOS ONE’s publication criteria as it currently stands. Therefore, we invite you to submit a revised version of the manuscript that addresses the points raised during the review process.

We look forward to receiving your revised manuscript.

Kind regards,

Hong-Liang Zhang, M.D., Ph.D.

Academic Editor

PLOS ONE

Journal Requirements:

Reviewers' comments:

Reviewer's Responses to Questions

**Comments to the Author**

Reviewer #1: All comments have been addressed

Reviewer #2: All comments have been addressed

2. Is the manuscript technically sound, and do the data support the conclusions?

Reviewer #1: Yes

Reviewer #2: Yes

3. Has the statistical analysis been performed appropriately and rigorously?

Reviewer #1: Yes

Reviewer #2: Yes

4. Have the authors made all data underlying the findings in their manuscript fully available?

Reviewer #1: Yes

Reviewer #2: Yes

5. Is the manuscript presented in an intelligible fashion and written in standard English?

Reviewer #1: No

Reviewer #2: Yes

Reviewer #1: Much morei improved, legible and able to discern the quality points made by the authors. There are a few grammatical errors noted eg. Page 22 "di not confirm this finding

I would recommend adding quantifying data when comparing how simliar or different the disorders are in certain test. For example it was noted central cerebrovascular flow was more diminished in COVID patients vs hED. Adding by how much etc. I find the discussion was lacking the actual quantifying data

Also recommend adding smaller tables that shows the highlighted points in the tests with the concurrent P Values and Confidence intervals rather than relying on the large table at the end.

Reviewer #2: All comments have been perfectly addressed. The manuscript is ready for publication. I also recommend highlight this article on the homepage of Plos One

**Do you want your identity to be public for this peer review?** For information about this choice, including consent withdrawal, please see our Privacy Policy

Reviewer #1: No

Reviewer #2: **Yes:** Shuo Yang

---

## [Author Response · Author response to Decision Letter 2]

11 Nov 2025

Dear Editor

We would like to express our sincere gratitude to both reviewers for their constructive feedback. We have carefully revised the paper in accordance with the suggestions. Below, we provide detailed responses to each of the reviewers' comments:

For authors

Peter Novak

Reviewer #1: Much morei improved, legible and able to discern the quality points made by the authors. There are a few grammatical errors noted eg. Page 22 "di not confirm this finding

I would recommend adding quantifying data when comparing how simliar or different the disorders are in certain test. For example it was noted central cerebrovascular flow was more diminished in COVID patients vs hED. Adding by how much etc. I find the discussion was lacking the actual quantifying data

Also recommend adding smaller tables that shows the highlighted points in the tests with the concurrent P Values and Confidence intervals rather than relying on the large table at the end.

Answer: Thank you for your suggestions. We added the Table 6 that summarizes the differences between Long COVID and ME/CFS. In the table 6 we focus in comparisons between Long COVID and ME/CFS because that was the main topic of the study. For that reasons we did not include Controls and hEDS in the table 6. We also added the difference in the discussion, for example to the sentence:

Both our patient groups exceeded that level of decline.

was expanded to:

Both our patient groups exceeded that level of decline (Long COVID –25% and ME/CFS -22%).

The sentence:

Our study detected frequent autonomic failure in both Long COVID and ME/CFS .

Was expanded to:

Our study detected frequent autonomic failure in both Long COVID (95%) and ME/CFS (89%).

For consistency, we prefer to use the sd (standard deviations) and not confidence intervals because we use sd’s in all other tables. Both sd’s and confidence intervals, together with p values are good measures of data variability.

We also added the paragraph discussing the central sensitization since we believe, this is important finding, particularly in Long COVID patients.

In addition, we corrected several typos and changed legends in figures (PASC was replaced with Long COVID).

---

## [Editor Report · Decision Letter 2]

14 Dec 2025

Dear Dr. Novak,

Thank you for submitting your manuscript to PLOS ONE. After careful consideration, we feel that it has merit but does not fully meet PLOS ONE’s publication criteria as it currently stands. Therefore, we invite you to submit a revised version of the manuscript that addresses the points raised during the review process.

We look forward to receiving your revised manuscript.

Kind regards,

Hong-Liang Zhang, M.D., Ph.D.

Academic Editor

PLOS One

Journal Requirements:

Additional Editor Comments:

Some minor changes are needed.

In Section 19 (Discussion - Central sensitization), "increase responsivness" has a misspelled noun. The correct spelling is "responsiveness" (with a double "s"). This term is critical to defining "central sensitization" (a core concept in the manuscript), so misspelling it weakens the clarity of the scientific definition.

In Section 19 (Discussion - Central sensitization), "central censitization" is misspelled. The correct term is "central sensitization". This error repeats the misspelling of a key pathophysiological concept, which may confuse readers (e.g., researchers focusing on pain/fatigue syndromes) and undermines the manuscript’s scientific accuracy.

In the "Materials and methods - Standard protocol approvals..." section, the sentence "the consent form signature was waived and authors of the study had access to information that could identify individual participants during data collection" contains a comma splice. Two independent clauses ("the consent form signature was waived" and "authors of the study had access...") are incorrectly joined by a single comma. To fix this, replace the comma with a period or a semicolon: "the consent form signature was waived. Authors of the study had access to information that could identify individual participants during data collection." This error compromises the grammatical flow of the ethics statement, a key section for ensuring research compliance.

In Section 7 (Inclusion and exclusion criteria), the sentence "Long COVID diagnosis was based on the following: 1) Evidence of previous SARS-CoV-2 infection established by a history of acute illness characterized by fever, cough and malaise confirmed by a positive SARS-CoV-2 infection, either by antigen test or polymerase chain reaction" has a misplaced modifier. The phrase "confirmed by a positive SARS-CoV-2 infection" incorrectly modifies "malaise" (a symptom) instead of "a history of acute illness" (the evidence of infection). Revise to: "Long COVID diagnosis was based on the following: 1) Evidence of previous SARS-CoV-2 infection—established by a history of acute illness (characterized by fever, cough, and malaise) and confirmed by a positive SARS-CoV-2 test (either antigen or polymerase chain reaction)." This correction clarifies the logical relationship between the illness history and diagnostic testing, avoiding misinterpretation of how the diagnosis was confirmed.

In Section 9 (Patient Reported Surveys), the sentence "The cutoff point >7 in the SAS score was considered to be clinically significant" lacks a definite article before "cutoff point". Since "cutoff point" refers to a specific threshold (for the Survey of Autonomic Symptoms), it should be "The cutoff point of >7" or "A cutoff point of >7". The current phrasing is grammatically incomplete and imprecise, as it does not clearly link the numerical value (>7) to the cutoff.

In Section 16 (Results - Symptoms), the abbreviation "ME/CSF" is used incorrectly. The correct abbreviation for "myalgic encephalomyelitis/chronic fatigue syndrome" is "ME/CFS" (with "CFS" instead of "CSF"—"CSF" refers to "cerebrospinal fluid", an unrelated biological fluid). This error appears multiple times in the Results section (e.g., "Long COVID and ME/CSF had a similar degree of complaints...") and creates critical confusion between the study’s core disorder (ME/CFS) and a distinct biological sample (CSF).

In Section 18 (Results - Invasive cardiopulmonary exercise testing), the sentence "Unadjusted resting stroke volume (p=0.01), exercise stroke volume (p=0.01), cardiac output (p=0.003), and oxygen uptake (p=0.001) were higher in Long COVID, but the differences were not significant after adjusting for BMI, which was higher in Long COVID" has a lack of parallel structure in parenthetical expressions. The p-values are presented as "(p=0.01)" but lack clarity on whether they refer to group comparisons. Revise to: "Unadjusted resting stroke volume (Long COVID vs. ME/CFS: p=0.01), exercise stroke volume (p=0.01), cardiac output (p=0.003), and oxygen uptake (p=0.001) were higher in Long COVID; however, these differences were no longer significant after adjusting for BMI (which was higher in Long COVID)." This improves grammatical parallelism and clarifies the context of the statistical tests.

In the "Response to Reviewers - Answer" section, the sentence "In the table 6 we focus in comparisons between Long COVID and ME/CFS because that was the main topic of the study" contains two preposition errors. First, "In the table 6" should be "In Table 6" (no article before numbered tables in academic writing); second, "focus in comparisons" should be "focus on comparisons" (the

---

## [Author Response · Author response to Decision Letter 3]

16 Dec 2025

Dear Editor

We would like to express our sincere gratitude for constructive feedback. We have carefully revised the paper in accordance with the suggestions. Below, we provide detailed responses to each of the reviewers' comments:

For authors

Peter Novak

Additional Editor Comments:

Some minor changes are needed.

In Section 19 (Discussion - Central sensitization), "increase responsivness" has a misspelled noun. The correct spelling is "responsiveness" (with a double "s"). This term is critical to defining "central sensitization" (a core concept in the manuscript), so misspelling it weakens the clarity of the scientific definition.

Answer: the typo in the word “responsiveness” was corrected.

In Section 19 (Discussion - Central sensitization), "central censitization" is misspelled. The correct term is "central sensitization". This error repeats the misspelling of a key pathophysiological concept, which may confuse readers (e.g., researchers focusing on pain/fatigue syndromes) and undermines the manuscript’s scientific accuracy.

Answer: the typo in the word “sensitization” was corrected.

In the "Materials and methods - Standard protocol approvals..." section, the sentence "the consent form signature was waived and authors of the study had access to information that could identify individual participants during data collection" contains a comma splice. Two independent clauses ("the consent form signature was waived" and "authors of the study had access...") are incorrectly joined by a single comma. To fix this, replace the comma with a period or a semicolon: "the consent form signature was waived. Authors of the study had access to information that could identify individual participants during data collection." This error compromises the grammatical flow of the ethics statement, a key section for ensuring research compliance.

Answer: The sentence “The study…” was split to two sentences as follows: The study was approved by the Institutional Review Board of the Brigham and Women’s Hospital, Harvard University, as a minimal-risk study, and the consent form signature was waived. Authors of the study had access to information that could identify individual participants during data collection.

In Section 7 (Inclusion and exclusion criteria), the sentence "Long COVID diagnosis was based on the following: 1) Evidence of previous SARS-CoV-2 infection established by a history of acute illness characterized by fever, cough and malaise confirmed by a positive SARS-CoV-2 infection, either by antigen test or polymerase chain reaction" has a misplaced modifier. The phrase "confirmed by a positive SARS-CoV-2 infection" incorrectly modifies "malaise" (a symptom) instead of "a history of acute illness" (the evidence of infection). Revise to: "Long COVID diagnosis was based on the following: 1) Evidence of previous SARS-CoV-2 infection—established by a history of acute illness (characterized by fever, cough, and malaise) and confirmed by a positive SARS-CoV-2 test (either antigen or polymerase chain reaction)." This correction clarifies the logical relationship between the illness history and diagnostic testing, avoiding misinterpretation of how the diagnosis was confirmed.

Answer: The sentence “Long COVID…” was revised as recommended.

In Section 9 (Patient Reported Surveys), the sentence "The cutoff point >7 in the SAS score was considered to be clinically significant" lacks a definite article before "cutoff point". Since "cutoff point" refers to a specific threshold (for the Survey of Autonomic Symptoms), it should be "The cutoff point of >7" or "A cutoff point of >7". The current phrasing is grammatically incomplete and imprecise, as it does not clearly link the numerical value (>7) to the cutoff.

Answer: The article “The” was placed before “cutoff point”

In Section 16 (Results - Symptoms), the abbreviation "ME/CSF" is used incorrectly. The correct abbreviation for "myalgic encephalomyelitis/chronic fatigue syndrome" is "ME/CFS" (with "CFS" instead of "CSF"—"CSF" refers to "cerebrospinal fluid", an unrelated biological fluid). This error appears multiple times in the Results section (e.g., "Long COVID and ME/CSF had a similar degree of complaints...") and creates critical confusion between the study’s core disorder (ME/CFS) and a distinct biological sample (CSF).

Answer: The words “ME/CSF” were corrected with “ME/CFS”

In Section 18 (Results - Invasive cardiopulmonary exercise testing), the sentence "Unadjusted resting stroke volume (p=0.01), exercise stroke volume (p=0.01), cardiac output (p=0.003), and oxygen uptake (p=0.001) were higher in Long COVID, but the differences were not significant after adjusting for BMI, which was higher in Long COVID" has a lack of parallel structure in parenthetical expressions. The p-values are presented as "(p=0.01)" but lack clarity on whether they refer to group comparisons. Revise to: "Unadjusted resting stroke volume (Long COVID vs. ME/CFS: p=0.01), exercise stroke volume (p=0.01), cardiac output (p=0.003), and oxygen uptake (p=0.001) were higher in Long COVID; however, these differences were no longer significant after adjusting for BMI (which was higher in Long COVID)." This improves grammatical parallelism and clarifies the context of the statistical tests.

Answer: The sentence “Unadjusted..” was modified as suggested.

In the "Response to Reviewers - Answer" section, the sentence "In the table 6 we focus in comparisons between Long COVID and ME/CFS because that was the main topic of the study" contains two preposition errors. First, "In the table 6" should be "In Table 6" (no article before numbered tables in academic writing); second, "focus in comparisons" should be "focus on comparisons" (the

Answer: The preposition errors were corrected in the response to reviewers’ file.

---

## [Editor Report · Decision Letter 3]

5 Jan 2026

Shared Autonomic Phenotype of Long COVID and Myalgic Encephalomyelitis/Chronic Fatigue Syndrome

PONE-D-25-02329R3

Dear Dr. Novak,

We’re pleased to inform you that your manuscript has been judged scientifically suitable for publication and will be formally accepted for publication once it meets all outstanding technical requirements.

Kind regards,

Hong-Liang Zhang, M.D., Ph.D.

Academic Editor

PLOS One

Additional Editor Comments (optional):

The reviewers' concerns have been fully addressed.
---

## [Editor Report · Acceptance letter]

PONE-D-25-02329R3

PLOS One

Dear Dr. Novak,

I'm pleased to inform you that your manuscript has been deemed suitable for publication in PLOS One. Congratulations! Your manuscript is now being handed over to our production team.

Kind regards,

on behalf of

Dr. Hong-Liang Zhang

Academic Editor

PLOS One